# Overlooked and widespread pennate diatom-diazotroph symbioses in the sea

Christopher R. Schvarcz [1✉], Samuel T. Wilson [1], Mathieu Caffin[1], Rosalina Stancheva [2], Qian Li [1,4], Kendra A. Turk-Kubo [3], Angelicque E. White [1], David M. Karl [1], Jonathan P. Zehr[3] & Grieg F. Steward [1]

Persistent nitrogen depletion in sunlit open ocean waters provides a favorable ecological niche for nitrogen-fixing (diazotrophic) cyanobacteria, some of which associate symbiotically with eukaryotic algae. All known marine examples of these symbioses have involved either centric diatom or haptophyte hosts. We report here the discovery and characterization of two distinct marine pennate diatom-diazotroph symbioses, which until now had only been observed in freshwater environments. Rhopalodiaceae diatoms *Epithemia pelagica* sp. nov. and *Epithemia catenata* sp. nov. were isolated repeatedly from the subtropical North Pacific Ocean, and analysis of sequence libraries reveals a global distribution. These symbioses likely escaped attention because the endosymbionts lack fluorescent photopigments, have *nifH* gene sequences similar to those of free-living unicellular cyanobacteria, and are lost in nitrogen-replete medium. Marine Rhopalodiaceae-diazotroph symbioses are a previously overlooked but widespread source of bioavailable nitrogen in marine habitats and provide new, easily cultured model organisms for the study of organelle evolution.

[1] Department of Oceanography, Daniel K. Inouye Center for Microbial Oceanography: Research and Education (C-MORE), University of Hawaiʻi at Mānoa, Honolulu, HI 96822, USA. [2] Department of Biological Sciences, California State University San Marcos, San Marcos, CA 92096, USA. [3] Department of Ocean Sciences, University of California, Santa Cruz, CA 95064, USA. [4] Present address: School of Oceanography, Shanghai Jiao Tong University, Shanghai, China. ✉email: schvarcz@hawaii.edu

Biological dinitrogen ($N_2$) fixation is a globally important process supporting primary production in terrestrial and aquatic ecosystems[1]. In the nitrogen (N)-depleted surface waters of open ocean biomes, up to ~50% of new production can be supported by $N_2$ fixation[2]. Identifying the spectrum of $N_2$-fixing plankton and their diverse physiologies is essential for predicting the effects of future ocean warming on biological productivity[3]. Although the intrinsic biochemical capacity to fix $N_2$ is restricted to bacteria and archaea, some eukaryotes have adapted to low-nitrogen concentrations in the oligotrophic ocean by establishing mutualistic symbioses with diazotrophic bacteria[4]. The two most prominent types of marine diazotroph symbioses described thus far are the facultative associations between heterocyst-forming cyanobacteria and centric diatoms[5,6] and the association between unicellular cyanobacteria and haptophytes[7,8].

An endosymbiotic relationship between unicellular diazotrophic cyanobacteria and pennate diatoms in the family Rhopalodiaceae has been identified in freshwater and brackish environments but has never been reported in the ocean[9]. Freshwater species in the genera *Epithemia* Kützing and *Rhopalodia* O. Müller host obligate, $N_2$-fixing endosymbionts of cyanobacterial origin, also referred to as 'spheroid bodies'[10–12]. *Epithemia*-spheroid body symbiosis is a valuable model system for studying the transformation of endosymbionts into organelles[13,14]. For example, endosymbionts in different species of freshwater Rhopalodiaceae differ in the genes and pathways retained after reductive evolution, reflecting differences in the trajectories of their organellogenesis[15].

In this work, we report the discovery of two species of oceanic, endosymbiont-bearing, rhopalodiacean diatoms, *Epithemia pelagica* Schvarcz, Stancheva & Steward sp. nov. and *Epithemia catenata* Schvarcz, Stancheva & Steward sp. nov. We show that *Epithemia* symbionts are globally distributed in the marine environment and that *E. pelagica* and *E. catenata* symbioses exhibit unique daily patterns of $N_2$ fixation.

## Results and discussion

The *Epithemia* strains were isolated from seawater samples collected from the subtropical North Pacific Ocean at Station ALOHA (22°45' N, 158°00' W)[16]. Samples were collected throughout the year and from multiple depths extending through the euphotic zone (Supplementary Table 1). Overall, seven strains were isolated that represent two morphologically and genetically distinct species. *E. pelagica* is characterized by small solitary cells, asymmetrical along the apical axis (Fig. 1a–c, n), while *E. catenata* cells are larger, nearly symmetric along the apical axis, hyaline, and chain-forming (Fig. 1d–g, o). Both species typically foster 1–2 unicellular endosymbionts per cell, which have average cell dimensions of $2.9 \times 2.4$ μm for *E. pelagica* (Fig. 1c, h, n) and $4.0 \times 2.5$ μm for *E. catenata* (Fig. 1f, k, o) and tend to be centrally located next to the host cell's nucleus (Fig. 1n, o). Similar to freshwater Rhopalodiaceae[17], the endosymbionts of *E. pelagica* (*Ep*SB) and *E. catenata* (*Ec*SB) lack detectable chlorophyll and phycoerythrin (Fig. 1h–m), implying that these endosymbionts may have lost their ability to photosynthesize and therefore have an obligate requirement for fixed carbon from their host. The endosymbionts were never observed growing outside of host cells and the host cultures lost their endosymbionts after being propagated for extended periods on N-replete medium (K medium[18], with $5 \times 10^{-5}$ M $NH_4^+$, $8.82 \times 10^{-4}$ M $NO_3^-$; Supplementary Fig. 20). Furthermore, the hosts showed little to no growth when subsequently returned to low-N medium ($5 \times 10^{-8}$ M $NH_4^+$).

Phylogenetic analyses support the hypothesis that the endosymbionts are obligate and are coevolving with their *Epithemia* host. The endosymbiont SSU (encoding 16S rRNA) and *nifH* (encoding nitrogenase iron protein) gene sequences cluster by host species and are highly conserved among different strains of the same host (99.5–100% nucleotide sequence identity). Such partner fidelity is expected for obligate symbionts and is similarly observed in marine prymnesiophyte-UCYN-A symbioses[19,20]. The endosymbionts *Ep*SB and *Ec*SB reside on distinct phylogenetic branches, forming a monophyletic clade with the spheroid bodies of freshwater species *E. turgida* (Ehrenberg) Kützing (*Et*SB) and *Rhopalodia gibberula* (Ehrenberg) O. Müller (*Rg*SB) (Fig. 2b). The diatom host phylogeny places *E. pelagica* and *E. catenata* on separate branches of the revised *Epithemia* lineage[21], with their nearest neighbors being *Rhopalodia* sp. 13vi08.2B (GCCT21) and *Rhopalodia* sp. 21IV14-4D (voucher HK433), respectively (Fig. 2a). Both of these *Rhopalodia* spp. were isolated from coastal marine environments but were not noted to contain endosymbionts[21,22]. *E. catenata*'s cell morphology differs significantly from the rest of Rhopalodiales[21] but shares many characteristics with the tentatively classified *Nitzschia nienhuisii* F.A.S. Sterrenburg & F.J.G. Sterrenburg[23], for which no molecular data exist. Spheroid body-like structures are visible in a previously captured photomicrograph of *N. nienhuisii* (Figure 89 in Lobban[24]), suggesting that this diatom species, which has been observed in the Pacific[24] and Atlantic[23] Oceans and Caribbean Sea[25,26], may also harbor endosymbionts.

*E. pelagica*-like symbioses have a global distribution throughout tropical and subtropical oceans (Fig. 3). Nucleotide sequences 100% identical to the *nifH* gene of *Ep*SB were observed in samples from the North Pacific Ocean, North and South Atlantic Oceans, Indian Ocean, and the seas surrounding China, Philippines, and Japan. These include amplified *nifH* sequences from NCBI's non-redundant nucleotide (nt) and Sequence Read Archive (SRA) databases ($n = 81{,}023$; ave. length = 272 bp), as well as unamplified sequences from *Tara* Oceans metagenomes and metatranscriptomes corresponding to an assembled transcript (unigene MATOU-v1_93255274) that covers the entire length of our *Ep*SB *nifH* sequences (760 bp). MATOU-v1_93255274 was most frequently found in metagenomes and metatranscriptomes generated from eukaryote-associated size fractions (2, 10, 8, and 2 samples corresponding to the 0.8–5 μm, 5–20 μm, 20–180 μm, and 180–2000 μm size fractions, respectively). *Epithemia* symbioses may be even more abundant and widespread, because lowering the threshold of nucleotide identity from 100% to 98% expands the global distribution of *Ep*SB-like *nifH* sequences to include the Gulf of Mexico, Coral Sea, and the Arctic Ocean (Supplementary Fig. 21a). *Ec*SB-like *nifH* sequences were rarer and only found in samples from the North Pacific Ocean and Coral Sea (Supplementary Fig. 21b). At Station ALOHA, where the isolates were obtained, quantitative PCR measurements of *E. pelagica* symbioses indicated host LSU and *Ep*SB *nifH* gene copies as high as $18 \pm 8 \times 10^3$ $L^{-1}$ and $0.7 \pm 0.2 \times 10^3$ $L^{-1}$, respectively (Supplementary Fig. 22a). Identical *Ep*SB *nifH* sequences were also detected in metagenomes constructed from sinking particles collected at 4000 m depth at Station ALOHA over a 3-year period (Supplementary Fig. 22b)[27,28]. The recurring presence of *Ep*SB sequences in sinking particles collected in the bathypelagic zone of the water column is strong evidence for a sustained population in the surface waters of the North Pacific Subtropical Gyre.

The daily patterns of $N_2$ fixation in *E. pelagica* and *E. catenata* endosymbionts are distinct from other pelagic diazotrophs. In general, $N_2$ fixation by marine cyanobacteria occurs during either the day or the night[29]. For both *Ep*SB and *Ec*SB, $N_2$ fixation occurred during the day and night (Fig. 4). During the day, $N_2$ fixation ceases either a few hours prior to the end of the light period (*E. pelagica*; Fig. 4a) or in conjunction with the lights being switched off (*E. catenata*; Fig. 4b). For both strains, $N_2$ fixation was subsequently undetected for the first 6 h of the dark

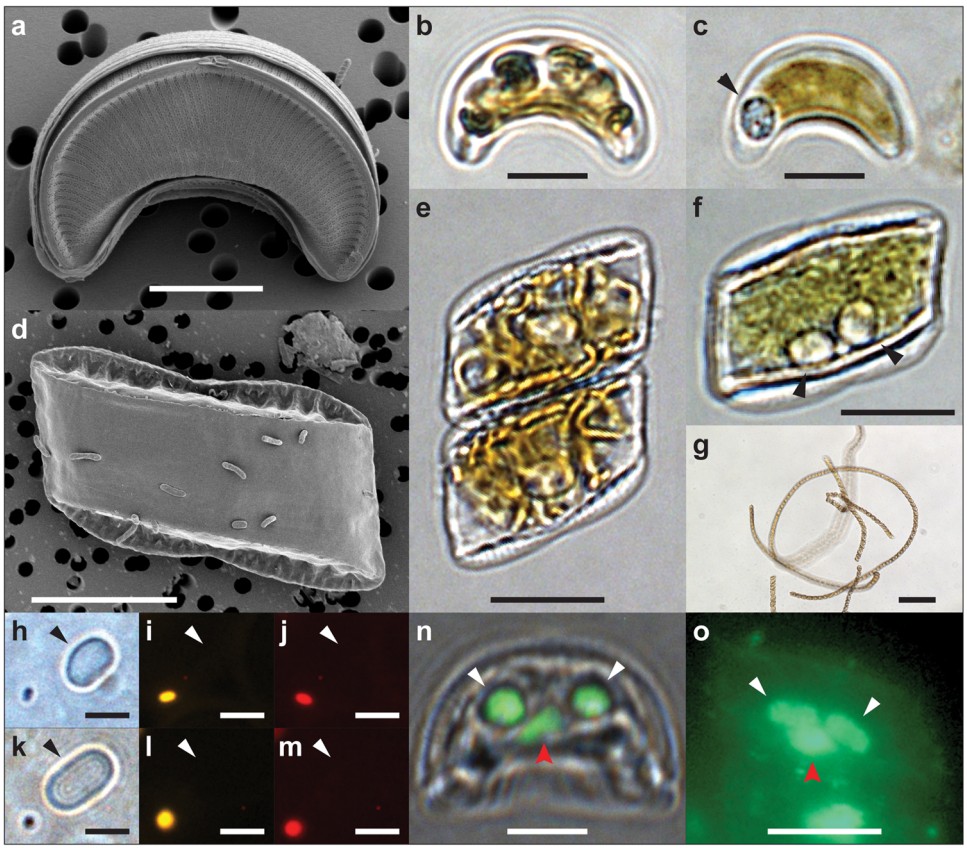

**Fig. 1 Characterization of marine *Epithemia* and their endosymbionts. a**, **b**, **c** Micrographs of *E. pelagica* UHM3200 in valve view, including visualization by **a** scanning electron microscopy, and light microscopy for **b** live cells and **c** cells osmotically shocked to display the endosymbionts, as indicated by the black arrow. The osmotic shock treatment (**c**, **f**) disrupts host cell contents and displaces the endosymbionts. The natural intracellular location of endosymbionts is illustrated in **n** and **o**. **d**, **e**, **f**, **g** Micrographs of *E. catenata* UHM3210 in girdle view, including visualization by **d** scanning electron microscopy, light microscopy for **e** live cells and **f** osmotically shocked cells with two endosymbionts highlighted by black arrowheads, and **g** undisturbed live cells growing in long chains. Micrographs of *Ep*SB (**h**, **i**, **j**) and *Ec*SB (**k**, **l**, **m**) released from crushed UHM3200 and UHM3210 host cells, respectively, as seen under brightfield (**h**, **k**), phycoerythrin fluorescence (**i**, **l**), and chlorophyll fluorescence (**j**, **m**). Arrowheads point to the endosymbiont cells, and a single *Synechococcus* WH7803 cell is present in each field to serve as a positive control for fluorescence. **n**, **o** Fluorescence micrographs of fixed *E. pelagica* and *E. catenata* cells, respectively, where endosymbionts (white arrows) and nuclei (red arrows) have been stained with nucleic acid-binding SYBR Gold dye (green). In **n**, the fluorescence channel has been overlaid on a brightfield micrograph. Micrograph scale bars are 5 μm (**a**–**c**, **n**), 10 μm (**d**–**f**, **o**), 100 μm (**g**), and 3 μm (**h**–**m**). The experiment assessing the autofluorescence of released endosymbionts (**h**–**m**) was performed once, while all other micrographs represent results that were consistently reproduced in multiple experiments.

period and resumed just after midnight. Overall, *Ep*SB and *Ec*SB are able to fix N$_2$ for a much longer period of time during a day-night cycle than other marine diazotrophs, especially the unicellular cyanobacterium *Crocosphaera subtropica* Mareš & J.R. Johansen[30,31], which is the closest free-living relative of rhopalodiacean endosymbionts (Fig. 2b). *C. subtropica* synthesizes its carbohydrates during the day and respires them at night to fuel N$_2$ fixation, while the evolutionary transition to an endosymbiont has enabled the *Ep*SB and *Ec*SB spheroid bodies to perform N$_2$ fixation during the day which is most likely fueled by metabolism of the host cell[14]. A similar evolutionary transition is hypothesized to have occurred for UCYN-A which fixes N$_2$ during the daytime[32].

Analysis of marine *Epithemia* endosymbiont *nifH* sequences suggests that these endosymbionts have been commonly misidentified in oceanic samples as free-living unicellular cyanobacteria within the UCYN-C group (Supplementary Fig. 23). UCYN-C is considered to represent free-living unicellular cyanobacteria closely related to *Crocosphaera* Zehr, Rachel A. Foster, Waterbury & E. Webb (including strains formerly identified as *Cyanothece* Komárek[33]), but published reports of UCYN-C have

employed *nifH* primers and probes with an identical match to *E. pelagica* (Supplementary Fig. 24)[34–36]. Other studies have detected the presence of spheroid body-like sequences in marine metatranscriptomes[37] and *Tara* Oceans samples[38], but until now there were no characterized marine rhopalodiacean symbioses to support these observations. With the observed distribution of *Ep*SB- and *Ec*SB-like sequences throughout tropical and subtropical oceans, *Epithemia* symbioses may be the most geographically widespread eukaryote–diazotroph association on the planet, occurring in freshwater, brackish, and marine environments. Establishing the spatial-temporal patterns of marine *Epithemia* will be essential for determining their contribution to the marine N cycle and the biological carbon pump[2]. Furthermore, the repeated isolation of endosymbiont-bearing *Epithemia* species demonstrates their ease of cultivation compared to other marine diazotroph-eukaryote symbioses, which are either uncultivated or difficult to maintain long-term in culture[4]. Thus, marine *Epithemia* are valuable new model systems for genomic and experimental investigations, because they can be used to unravel the interplay of host-symbiont physiologies and the genetic adaptations that maintain these relationships[39].

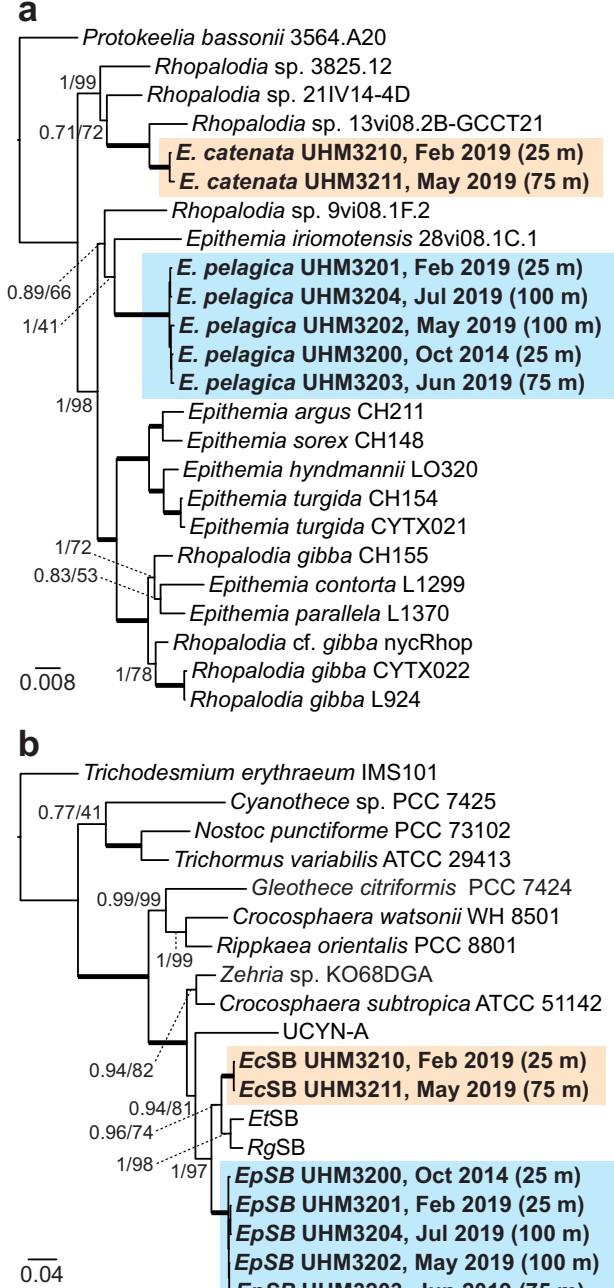

**a**

**b**

**Fig. 2 Phylogenetic analyses. a** Multigene phylogeny of the *E. pelagica* and *E. catenata* diatom hosts, based on the SSU (18S rRNA; 1322 nt), *psbC* (986 nt), and *rbcL* (1341 nt) genes. **b** Multigene phylogeny of the *Epithemia* endosymbionts, based on the SSU (16S rRNA; 420 nt) and *nifH* (760 nt) genes. The phylogenies are Bayesian majority consensus trees, and support values are provided for Bayesian and maximum likelihood methods (Bayesian posterior probabilities/ML bootstrap percent). Bold branches indicate complete support (posterior probability of 1 and bootstrap percent of 100), and the phylogeny scales are in units of nt substitutions per site. Labels for the new isolates (highlighted and bolded) include the dates and depths of collection. Accession numbers for all sequences are provided in the Source Data file.

**Systematic biology.** Phylum Bacillariophyta Karsten
  Class Bacillariophyceae Haeckel
    Order Rhopalodiales D. G. Mann
      Family Rhopalodiaceae (Karsten) Topachevs'kyj & Oksiyuk
        Genus *Epithemia* Kützing

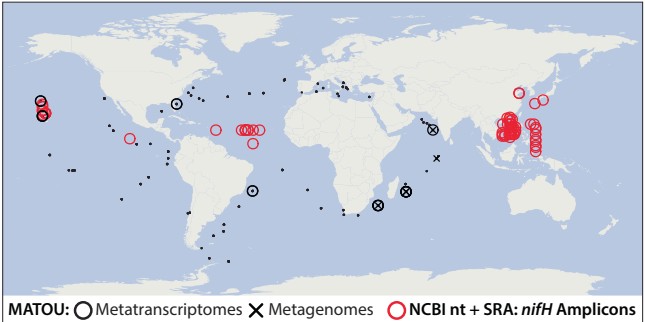

**Fig. 3 Global map of environmental sequences that share 100% nucleotide identity with an *E. pelagica nifH* phylotype.** Datasets screened include the Marine Atlas of *Tara* Oceans Unigenes (MATOU) and the representation of unigene MATOU-v1_93255274 in *Tara* Oceans metagenomes and metatranscriptomes, as well as *nifH* amplicons published in NCBI's non-redundant nucleotide (nt) and Sequence Read Archive (SRA) databases. Black dots indicate the location of *Tara* Oceans stations analyzed in MATOU. Source data are provided in the Source Data file.

*Epithemia pelagica* Schvarcz, Stancheva & Steward sp. nov.
Figure 1a–c, n; Supplementary Figs. 1a–n, 2a–h, 6–10.

**Holotype.** Slide UC2085162 from *Epithemia pelagica* UHM3201 deposited in the University and Jepson Herbaria at the University of California, Berkeley. Holotype specimen is illustrated in Supplementary Fig. 1b.

**Etymology.** The epithet refers to the pelagic habitat of this species, collected from the open Pacific Ocean.

**Type locality.** This species was isolated from seawater collected from a depth of 25 m in the North Pacific Subtropical Gyre (22°45′ N, 158°00′ W), Station ALOHA (water depth ca. 4800 m) on October 14, 2014 (*Epithemia pelagica* UHM3200) and February 21, 2019 (*Epithemia pelagica* UHM3201).

**Diagnosis.** This species is characterized by solitary, strongly dorsiventral, small cells, 6.7–17.8 μm long, 5–9.8 μm wide. Valves are lunate with rounded apices, convex dorsal margin, and concave ventral margin. Raphe-bearing keel is eccentric, positioned on the dorsal margin, slightly bent down towards the dorsal margin at the center of the valve. Transapical costae are fine and resolved only near the raphe keel, where they are internally thickened and function as fibulae beneath the raphe. Striae are not resolvable with light microscopy (LM). *E. pelagica* possesses all structural features of genus *Rhopalodia* (now *Epithemia*), but differs from other species by its minute size, weakly silicified frustules with delicate costae and very fine striae not resolvable with LM.

**Note.** This species also shares some morphological similarities with *Protokeelia* C.W. Reimer & J.J. Lee, such as minute size, lunate valve with undulate valve face, and protuberant raphe sternum.

**Description.** See Supplementary Note 1.

*Epithemia catenata* Schvarcz, Stancheva & Steward sp. nov.
Figure 1d–g, o; Supplementary Figs. 3a–c, 4a–k, 5a–f, 11, 12.

**Holotype.** Slide UC2085161 from *Epithemia catenata* UHM3210 deposited in the University and Jepson Herbaria at University of

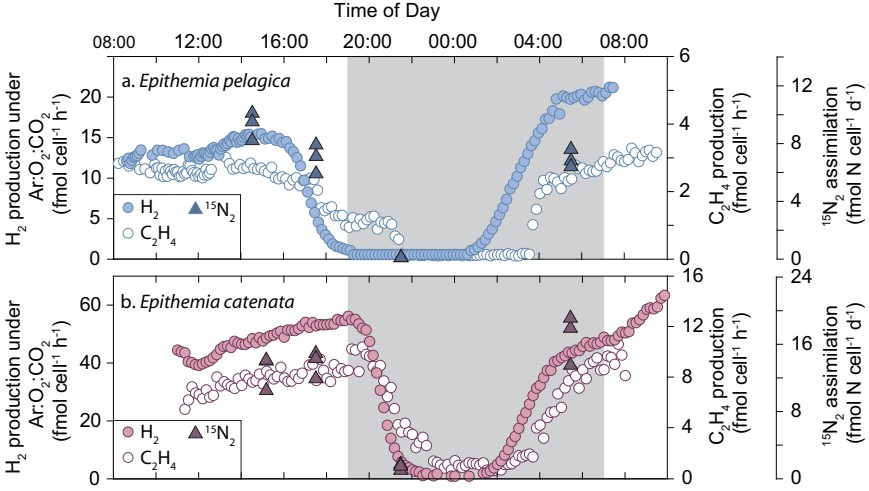

**Fig. 4 Daily patterns of N₂ fixation for *Epithemia*. a** Dihydrogen ($H_2$) production, ethylene ($C_2H_4$) production from acetylene ($C_2H_2$), and $^{15}N_2$ assimilation for *E. pelagica* UHM3200. **b** $H_2$ production, $C_2H_4$ production from $C_2H_2$, and $^{15}N_2$ assimilation for *E. catenata* UHM3210. Sampling resolution was 10 min for $H_2$ and $C_2H_4$ production and 2 h for the $^{15}N_2$ assimilation with the values represented by average ± standard deviation ($n = 3$). The $^{15}N_2$ assimilation beginning at 20:30 had rates of 0.1 fmol N cell$^{-1}$ d$^{-1}$ for *E. pelagica*. The night period is represented by gray shading in both plots. Source data are provided in the Source Data file.

California, Berkeley. Holotype specimen is illustrated in Supplementary Fig. 3a.

**Etymology**. The epithet refers to the colonial habit of this diatom, which forms long chains.

**Type locality**. This species was isolated from seawater collected from a depth of 25 m in the North Pacific Subtropical Gyre (22°45' N, 158°00' W), Station ALOHA (water depth ca. 4,800 m) on February 21, 2019 (*Epithemia catenata* UHM3210) and from a depth of 75 m on May 4th, 2019 (*Epithemia catenata* UHM3211).

**Diagnosis**. This species is characterized by cells joined together in chains. Frustules are delicate, translucent without visible striation, with wide, slightly rhomboidal girdles and narrower elliptical valves. The raphe keel is central or nearly so on valve face, continuing from pole to pole, slightly sigmoid, and fibulate, involved in cell-cell interlock. Frustules are 17.2–28.8 μm along apical axis, 8.2–13.7 μm along transapical axis, 11.7–16.9 μm along pervalvar axis, fibulae are 8–12 in 10 μm, 2–3 μm long. This species differs morphologically from all known *Epithemia/Rhopalodia* taxa by its colonial habit, raphe position, and frustular ultrastructure.

**Note**. The generic placement of *E. catenata* is based on current molecular phylogenetics data (Fig. 2a, Supplementary Figs. 13–19) and the presence of cyanobacterial endosymbionts. SH and AU tests of constrained phylogenetic topologies (Supplementary Table 3) show greater support for the inclusion of *E. catenata* within the genus *Epithemia*, while showing weaker support for the exclusion of *E. catenata* from either the genus *Epithemia* or family Rhopalodiaceae. *E. catenata* shares more morphological characteristics with *Nitzschia nienhuisii* than with other *Epithemia/Rhopalodia* species, such as gross frustule symmetry, keel structure and position, chain-formation, and hyaline frustules (LM Figs. 2–4 in Sterrenburg & Sterrenburg[23]; LM Figs. 89–92, SEM Figs. 93, 94 in Lobban[24]; SEM Fig. 2a, b, LM Fig. 2c, d in Lópes-Fuerte et al.[26]), suggesting that these two species may be congeneric. The initial description of *N. nienhuisii* acknowledges there is doubt with respect to *N. nienhuisii*'s generic ranking[23], and a later study employing SEM noted the

baffling structure of *N. nienhuisii*, which does not appear to possess fibulae on the keel typical of *Nitzschia* Hassall[24].

**Description**. See Supplementary Note 2.

## Methods

***Epithemia* isolation and culture**. The *Epithemia* cells were isolated from 0.5 L of seawater collected from depths of 25, 75, and 100 m in the North Pacific Subtropical Gyre (22°45' N, 158°00' W). Seawater was collected during the near-monthly Hawaii Ocean Time-series (HOT) expeditions to the long-term monitoring site Station ALOHA (water depth ca. 4800 m) in October 2014 (HOT cruise #266) and February–July 2019 (HOT cruises #310–313). Serial dilution (unialgal strains UHM3202, UHM3203, UHM3204) or micropipette isolation of single cells (clonal strains UHM3200, UHM3201, UHM3210, UHM3211) were used to establish the *Epithemia* cultures, which were grown in a seawater-based, low-nitrogen medium. Filtered (0.2 μm) and autoclaved, undiluted Station ALOHA seawater was amended with 2 μM EDTA, 50 nM ferric ammonium citrate, 7.5 μM phosphoric acid, trace metals (100 nM $MnSO_4$, 10 nM $ZnCl_2$, 10 nM $Na_2MoO_4$, 1 nM $CoCl_2$, 1 nM $NiCl_2$, 1 nM $Na_2SeO_3$), vitamins (50 μg/L inositol, 10 μg/L calcium pantothenate, 10 μg/L thiamin, 5 μg/L pyridoxine HCl, 5 μg/L nicotinic acid, 0.5 μg/L para-aminobenzoic acid, 0.1 μg/L folic acid, 0.05 μg/L biotin, 0.05 μg/L vitamin $B_{12}$), and 106 μM $Na_2SiO_3$. Although not tested here, simpler formulations of diazotroph media such as PMP[40] or RMP[41] may also be suitable for growing *Epithemia*, when made with 100% seawater and adding $Na_2SiO_3$. The cultures were subsequently incubated at 24 °C on a 12:12 h light:dark cycle with 50–100 μmol quanta m$^{-2}$ s$^{-1}$ using cool white fluorescent bulbs. All *E. pelagica* and *E. catenata* symbioses were stable under these medium and incubation conditions. *E. pelagica* was successfully isolated from at least one of the three depths that were targeted during each sampling occasion.

**Morphological observations**. *Epithemia* living and fixed cells were imaged by light and epifluorescence microscopy using a Nikon Eclipse 90i microscope at 40×–60× magnification. Diatom cell sizes were determined using >60 live, exponentially growing cells, imaged in either valve view (*E. pelagica*) or girdle view (*E. catenata*). Endosymbiont (spheroid body) cell sizes were averaged from DNA-stained cells for *E. pelagica* UHM3200 ($n = 78$) and *E. catenata* UHM3210 ($n = 91$), imaged by epifluorescence microscopy after preparing samples as follows: *Epithemia* cells were fixed in 4% glutaraldehyde for 30 min, pelleted at 1000 × g for 1 min, the supernatant was exchanged with 0.5% Triton X-100 (in autoclaved filtered seawater), samples were incubated for 10 min with gentle agitation, cells were then pelleted at 4000 × g for 1 min, supernatant was exchanged with autoclaved filtered seawater and fixed in 4% glutaraldehyde, and samples were stained with 1× final concentration of SYBR Gold nucleic acid stain (Invitrogen, cat. # S11494) for 2 h. For routine observations of endosymbionts (e.g., determining presence/absence and number per host cell), osmotic shock was used to disrupt the cell contents of diatom host cells and improve visualization of the endosymbionts. This was achieved by gently pelleting cells and exchanging the medium with either ultrapure water or 2–3 M NaCl solution, followed by immediate observation. While this is a simple technique for detecting and visualizing endosymbionts (Fig. 1c, f), it

does not accurately represent the natural location of endosymbionts within the host cells, as seen when compared to fixed cell preparations for epifluorescence microscopy (Fig. 1n, o). To assess the presence of fluorescent photopigments in endosymbiont cells, live host cells were pelleted at $4000 \times g$ for 5 min and crushed using a microcentrifuge tube pestle (SP Bel-Art, cat. # F19923-0000) to release the endosymbionts. The crushed pellet was resuspended in 75% glycerol containing live *Synechococcus* WH7803 cells (positive control for fluorescence), and samples were observed by epifluorescence microscopy using filter cubes appropriate for observing phycoerythrin (EX: 551/10, BS: 560, EM: 595/30) and chlorophyll (EX: 480/30, BS: 505, EM: 600LP).

The loss of endosymbionts from *Epithemia* cultures (UHM3200 and UHM3210) was observed after propagating cells for four months in nitrogen-replete medium (K)[18], where approximately 5–10% of the culture was transferred to fresh medium about every two weeks. Observations were only made at the end of the four-month period. Endosymbionts were not observed growing freely in these cultures, and the absence of endosymbionts within host cells was confirmed by the failure to observe spheroid bodies by light microscopy after osmotic shock of the diatoms, as well as a failure to amplify the endosymbiont SSU (16S rRNA) and *nifH* genes from cellular DNA extracts. PCR reactions were performed in parallel with DNA extracts from control cultures (grown in low-nitrogen medium), using the same template DNA amount (10 ng) and PCR conditions (see methods for *Marker gene sequencing and phylogenetics*).

Ultrastructural observations by electron microscopy (EM) were conducted for *E. pelagica* UHM3200 and *E. catenata* UHM3210. EM preparations of diatoms typically involve the oxidative removal of organic matter to uncover the fine details of frustule ultrastructure. However, in the case of *E. catenata*, oxidatively cleaned cells lacked structural integrity, leading to collapsed frustules when dried and viewed by scanning EM (SEM). For this reason, both species were prepared for SEM with and without (Fig. 1a, d) the oxidative removal of organic matter, and cleaned *E. catenata* frustules were further analyzed by transmission EM (TEM). To remove organic matter, 100 mL of exponentially growing culture was pelleted by centrifugation at $1000 \times g$ for 10 min and resuspended in 30% $H_2O_2$. Cells were boiled in $H_2O_2$ for 1–2 h, followed by rinsing cells six times in ultrapure water by sequential centrifugation at $1000 \times g$ for 10 min and resuspension of cell pellets. Suspensions of the cleaned cells were dried on aluminum foil and mounted on aluminum stubs with double-sided copper tape. For some *E. catenata* SEM preparations, the cleaned frustules were dehydrated in an ethanol dilution series and exchanged into hexamethyldisilazane (HMDS) prior to drying on aluminum foil; this was to minimize the collapse of frustules resulting from drying. To prepare cells with organic matter intact, 25 mL of exponentially growing culture was mixed with an equal volume of fixative solution (5% glutaraldehyde, 0.2 M sodium cacodylate pH 7.2, 0.35 M sucrose, 10 mM $CaCl_2$) and incubated overnight at 4 °C. Cells were gently filtered onto a 13 mm diameter 1.2 μm pore size polycarbonate membrane filter (Isopore, Millipore Sigma), washed with 0.1 M sodium cacodylate buffer (pH 7.4, 0.35 M sucrose), fixed with 1% osmium tetroxide in 0.1 M sodium cacodylate (pH 7.4), dehydrated in a graded ethanol series, and critical point dried. Filters were mounted on aluminum stubs with double-sided conductive carbon tape. All SEM stubs were sputter coated with Au/Pd, prior to observing on a Hitachi S-4800 field emission scanning electron microscope at the University of Hawai'i at Mānoa (UHM) Biological Electron Microscope Facility (BEMF). Cleaned *E. catenata* cells were prepared for TEM by drying a drop of sample on a formvar/carbon-coated grid and observing on a Hitachi HT7700 transmission electron microscope at UHM BEMF.

Additional light microscopy of hydrogen-peroxide cleaned frustules was conducted for *E. pelagica* UHM3201 and *E. catenata* UHM3210. Samples were mounted in Naphrax (PhycoTech, Inc., cat. # P-Naphrax200) and observed at 100× using an Olympus BX41 Photomicroscope (Olympus America Inc., Center Valley, Pennsylvania) with differential interference contrast optics and an Olympus SC30 Digital Camera at California State University San Marcos.

A key to the strains used in each micrograph is provided in Supplementary Table 2.

**Marker gene sequencing and phylogenetics.** For each *Epithemia* strain, 25–50 mL of culture was pelleted at $4000 \times g$ for 10 min, and DNA was extracted from the pellet using the ZymoBIOMICS DNA Miniprep Kit (Zymo Research, cat. # D4300). Marker genes were amplified with the Expand High Fidelity PCR System (Roche, cat. # 4743733001), using conditions previously described for genes SSU encoding 18S rRNA (Euk328f/Euk329r)[42], LSU encoding 28S rRNA (D1R/D2C)[43], *rbcL* (rbcL66+/dp7−)[44,45], *psbC* (psbC+/psbC−)[44], and *cob* (Cob1f/Cob2r)[21]. For the endosymbionts, a partial sequence for the SSU (16S rRNA) gene was amplified using a primer set targeting unicellular cyanobacterial diazotrophs, CYA359F/ Nitro821R[46,47], and the *nifH* gene was amplified using new primers specific to the *nifH* of *Cyanothece*-like organisms, ESB-nifH-F (5'-TACGGAAAAGGCGGTA TCGG-3') and ESB-nifH-R (5'-CACCACCAAGRATACCGAAGTC-3'), with a 55 °C annealing temperature and 75 s extension time. All primers were synthesized by Integrated DNA Technologies (IDT). Amplified products were cloned and transformed into *E. coli* using the TOPO TA Cloning Kit for Sequencing (Invitrogen, cat. # K457501), and plated colonies were picked and grown in Circlegrow medium (MP Biomedicals, cat. # 113000132). Plasmids were extracted with the Zyppy Plasmid Miniprep kit (Zymo Research, cat. # D4019) and sequenced from

the M13 vector primers using Sanger technology at GENEWIZ (South Plainfield, NJ). For the diatom SSU (18S rRNA) gene, sequencing reactions were also performed using the 502f and 1174r primers[48].

Phylogenetic trees (Fig. 2) were inferred using concatenated alignments for both diatom host genes (SSU encoding 18S rRNA, *psbC*, *rbcL*) and endosymbiont genes (SSU encoding 16S rRNA, *nifH*). For each gene, nucleotide sequences were aligned using MAFFT v7.453[49] (L-INS-i method), and sites with gaps or missing data were removed. An appropriate nucleotide substitution model was selected for each gene alignment using jModelTest v2.1.10[50]. Bayesian majority consensus trees were inferred from the concatenated alignments using MrBayes v3.2.7[51] with two runs of 4–8 chains, until the average standard deviation of split frequencies dropped below 0.01. Maximum likelihood bootstrap values were generated for the Bayesian tree using RAxML v8.2.12[52], implemented with 1000 iterations of rapid bootstrapping. To further analyze the phylogenetic position of the new *Epithemia* species in the broader context of Surirellales and Rhopalodiales diatoms, individual gene trees (SSU encoding 18S rRNA, LSU, *rbcL*, *psbC*, and *cob*; Supplementary Figs. 13–19) were constructed from sequences aligned using MAFFT (automatic detection method) and trimmed using trimAl v1.2[53] (gappyout method). rRNA gene phylogenies were also inferred using sequences aligned according to the global SILVA alignment for SSU and LSU genes using SINA[54], which were either left untrimmed in the case of the LSU gene or trimmed to remove highly variable positions (SINA's "012345" positional variability filter) and gappy positions (trimAL v1.2, gappyout method) in the case of the SSU gene. These trimming strategies were selected based on their ability to maximize the monophyly of the previously described Rhopalodiales clade and minimize the separation of known conspecific strains, such as the strains of *E. pelagica* described here. All gene phylogenies were inferred using the Bayesian methods described above. To investigate the level of support for constrained tree topologies placing *E. catenata* within or outside of the genus *Epithemia* and family Rhopalodiaceae, SH[55] and AU[56] statistical tests were performed in IQ-TREE 2[57] (implementing ModelFinder[58]) using all alignments from the individual gene trees (Supplementary Table 3).

Given *E. catenata*'s unusual morphology, test trees were inferred with the inclusion of diatom sequences from orders Bacillariales (*Nitzschia*, *Pseudo-nitzschia*), Cymbellales (*Didymosphenia*), Naviculales (*Amphiprora*, *Navicula*, *Pinnularia*), and Thalassiophysales (*Amphora*, *Halamphora*, *Thalassiophysa*); however, *E. catenata* was consistently placed within Rhopalodiales, and these trees were not pursued further.

An additional *nifH* phylogeny was constructed using all environmental sequences from NCBI's non-redundant nucleotide (nt) database >300 bp and sharing >95% nucleotide sequence identity with *Ep*SB and *Ec*SB *nifH* sequences (Supplementary Fig. 23), including 51 environmental sequences from prior studies investigating marine diazotrophs[34,59–66]. Environmental *nifH* sequences were aligned to the previously generated *nifH* sequence alignment using MAFFT (automatic method detection and addfragments options), and the best-scoring maximum likelihood phylogeny was inferred using RAxML with 1000 iterations of rapid bootstrapping. NCBI accession numbers for all tree sequences are in the Source Data file.

**Analysis of *Epithemia* endosymbiont *nifH* sequences in environmental datasets.** Nucleotide sequences for *Ep*SB and *Ec*SB *nifH* were queried against NCBI's non-redundant nucleotide (nt) database using webBLAST[67] (megablast; https:// blast.ncbi.nlm.nih.gov/) and SRA databases for *nifH* amplicon sequencing projects from the marine environment using the SRA Toolkit[68] (dc-megablast, with database validation using vdb-validate; https://github.com/ncbi/sra-tools). Database hits with 98–100% nucleotide identity over an alignment of the entire subject sequence (BLAST alignment length = subject sequence length) were identified, and the associated sample's latitude and longitude coordinates (where available) were mapped. Coordinates were also mapped for metagenome and metatranscriptome samples containing matches to unigene MATOU-v1_93255274 from the Marine Atlas of *Tara* Oceans Unigenes[69], a unigene that shares 100% identity over the entire length of the *Ep*SB UHM3202 *nifH* sequence and >99.4% identity with all other *Ep*SB *nifH* sequences.

The presence of *Ep*SB and *Ec*SB *nifH* sequences was examined in metagenomes prepared from sinking particles collected at 4000 m depth at Station ALOHA[27,28]. The sinking particles were collected during intervals of 12, 10, and 8 days during 2014, 2015, and 2016, respectively, using a McLane sediment trap equipped with a 21-sample bottle carousel. The presence of *Ep*SB and *Ec*SB *nifH* sequences in the metagenomes was assessed by blastn[70], after first removing low quality bases from metagenomic reads using Trimmomatic v0.39[71] (parameters: LEADING:20 TRAILING:20 MINLEN:100). For each sediment trap metagenome, the total number of reads matching *Ep*SB or *Ec*SB *nifH* nucleotide sequences with 100% identity were tallied and normalized to the total number of reads in the database. Only *Ep*SB-matching reads were detected in this analysis.

**Quantitative PCR.** Specific PCR primers were designed targeting a 102 bp region of *E. pelagica*'s LSU gene (Epel-LSU-F, 5'-GAAACCAGTGCAAGCCAAC-3'; Epel-LSU-R, 5'-AGGCCATTATCATCCCTTGTC-3') and an 85 bp region *Ep*SB's *nifH* gene (EpSB-nifH-F, 5'-CACACTAAAGCACAAACTACC-3'; EpSB-nifH-R, 5'-CAAGTAGTACTTCGTCTAGCTC-3') and were synthesized by IDT. Gene copy concentrations were quantified for Station ALOHA water samples (~2 L) collected

by Niskin bottles at 5, 25, 45, 75, 100, 125, 150, and 175 m on January 16 and July 1 (except 5 m), 2014, during HOT cruises #259 and #264. Samples were filtered onto 25 mm diameter, 0.02 μm pore size aluminum oxide filters (Anotop; Whatman, cat. # WHA68092102) and stored at −80 °C until extracting DNA using the MasterPure Complete DNA and RNA Purification Kit (Epicentre, cat. # MC85200) according to Mueller et al.[72]. Briefly, a 3-mL syringe filled with 1 mL of tissue and cell lysis solution (MasterPure) containing 100 μg mL$^{-1}$ proteinase K was attached to the outlet of the filter, and the filter inlet was sealed with a second 3-mL syringe. The lysis solution was pulled halfway through to saturate the filter membrane, and the entire assembly was incubated at 65 °C for 15 min while attached to a rotisserie in a hybridization oven rotating at ca. 16 rpm. The lysis buffer was then drawn fully into the inlet syringe, transferred to a microcentrifuge tube, and placed on ice. The remaining steps for protein precipitation and removal and nucleic acid precipitation were carried out following the manufacturer's instructions. For each sample, DNA was resuspended in a final volume of 100 μL. Quantitative PCR (qPCR) was performed using the PowerTrack SYBR Green Master Mix system (Applied Biosystems, cat. # A46109) and run on an Eppendorf Mastercycler ep$gradient$ S realplex$^2$ real-time PCR machine. Reactions (20 μL total volume) were prepared according to the manufacturer's protocol, containing 500 nM of each primer. Sample reactions (four replicates) contained 2 μL of environmental DNA extract (24–76 ng DNA), while standards (three replicates) contained 2 μL of gBlocks Gene Fragments (IDT) that were prepared at 1, 2, 3, 4, 5, and 6 log gene copies/μL. The gBlocks Gene Fragments were 500 bp in length and encompassed the entire *E. pelagica* UHM3201 LSU sequence and positions 1–500 of the *Ep*SB UHM3201 *nifH* sequence, respectively. The main cycling conditions consisted of an initial denaturation and enzyme activation step of 95 °C for 2 min, followed by 40 cycles of 95 °C for 5 s and 57 °C or 55 °C for 30 s for the LSU and *nifH* genes, respectively. Melting curves were analyzed to verify the specificity of the amplifications, and reactions containing *Epithemia catenata* DNA extract were included as negative controls. Reaction efficiencies were 104.23% and 95.15% for the LSU and *nifH* genes, respectively. The limit of detection for these assays was not empirically determined. gBlocks sequences, qPCR threshold cycle values, and conversion equations are provided in the Source Data file.

**Physiology experiments**. The daily patterns of N$_2$ fixation were quantified for *E. pelagica* UHM3200 and *E. catenata* UHM3210 using two techniques: acetylene (C$_2$H$_2$) reduction to ethylene (C$_2$H$_4$) and argon induced dihydrogen (H$_2$) production (AIHP). Both analyses were conducted using a gaseous flow-through system that quantified the relevant trace gas on the sample outlet line with a temporal resolution of 10 min[73]. To conduct the measurements, a 10-mL subsample of each *Epithemia* culture was placed in a 20-mL borosilicate vial and closed using gas-tight rubber stoppers and crimp seals. Separate bottles were used for H$_2$ production and C$_2$H$_2$ reduction. During the experimental period, the temperature was maintained at 25 ± 0.2 °C using a benchtop incubator (Incu-Shaker; Benchmark Scientific) and light exposure was 200 μmol photons m$^{-2}$ s$^{-1}$ at wavelengths of 380–780 nm with a 12:12 h square light:dark cycle (Prime HD+; Aqua Illumination). To conduct the AIHP method, the sample vial containing the culture was flushed with a high purity gas mixture consisting of argon (makeup gas; 80%), oxygen (20%), and carbon dioxide (0.04%). In the absence of N$_2$, all of the electrons that would have been used to reduce N$_2$ to NH$_3$ are diverted to H$_2$ production, thereby providing a measure of Total Nitrogenase Activity (TNA). The C$_2$H$_2$ reduction assay also represents a measure of TNA. Our analytical set-up introduced C$_2$H$_2$ at a 1% addition (vol/vol) to the high purity air with a total flow rate (13 mL min$^{-1}$) identical to the AIHP method. The gas emissions were analyzed using separate reductive trace gas analyzers that were optimized for the quantification of H$_2$ and C$_2$H$_4$. To verify the observed daily patterns in N$_2$ fixation, $^{15}$N$_2$ assimilation measurements were conducted on triplicate samples of *Epithemia* cultures at targeted time points. Five milliliters of $^{15}$N-enriched seawater was added to the subsamples, which were subsequently crimp sealed and incubated for a 2 h period with the same light and temperature conditions as the daily gas measurements. At the end of the incubation, the contents of each vial were filtered onto a pre-combusted glass fiber filter. The concentration and isotopic composition (δ$^{15}$N) of particulate nitrogen for incubated and non-incubated (i.e., natural abundance) samples was measured using an elemental analyzer/isotope ratio mass spectrometer (Carlo-Erba EA NC2500 coupled with a ThermoFinnigan Delta Plus XP). For each of the described analyses, cell-specific rates were calculated based on the average of triplicate cell concentration measurements, obtained from cell samples preserved at 4 °C with Lugol's iodine solution and quantified within a week using a Sedgwick-Rafter counting chamber (Electron Microscopy Sciences, cat. # 68050-52). All rate measurement data is provided in the Source Data file.

**Reporting summary**. Further information on research design is available in the Nature Research Reporting Summary linked to this article.

## Data availability
Sequences produced for this study have been deposited in GenBank under accession numbers MW562846–MW562894. Analyses were also conducted using data from NCBI's non-redundant nucleotide (nt) and protein (nr) databases and the following SRA run database accessions: DRR075654–DRR075675, DRR090493–DRR090512,

DRR119299–DRR119318, SRR11748760–SRR11748769, SRR11784070–SRR11784101, SRR1994968–SRR1994982, SRR2846720, SRR2846725, SRR2848263–SRR2848264, SRR2848267, SRR2849323, SRR2849339, SRR2849358, SRR2849373, SRR2849384, SRR2849398, SRR2976582–SRR2976583, SRR2988260, SRR3225470–SRR3225471, SRR3275263–SRR3275264, SRR3502230, SRR3502520–SRR3502528, SRR3898627–SRR3898675, SRR3924383–SRR3924409, SRR5083564–SRR5083575, SRR5693565–SRR5693584, SRR5693645–SRR5693657, SRR576444, SRR576446, SRR576451, SRR576453–SRR576469, SRR5814033–SRR5814187, SRR6057892–SRR6057916, SRR6299285–SRR6299287, SRR6334371–SRR6334373, SRR7142301–SRR7142368, SRR7239923–SRR7239946, SRR7527146–SRR7527159, SRR7632639–SRR7632648, SRR7632653, SRR7632671–SRR7632680, SRR7648270, SRR7648273–SRR7648274, SRR7648284–SRR7648299, SRR7648310, SRR7648320–SRR7648321, SRR7648326–SRR7648327, SRR7648331–SRR7648339, SRR7648341, SRR7648343, SRR7648345–SRR7648350, SRR7668191, SRR7699187–SRR7699216, SRR8104593–SRR8104721, SRR8247196–SRR8247211, SRR8468235–SRR8468237, SRR8468246–SRR8468261, SRR8844064–SRR8844199, SRR9675236. Source data for Figs. 2–4 and Supplementary Figs. 13–17, 18e, 20, and 21 are provided with this paper.

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

## Acknowledgements

This work was supported by National Science Foundation awards (OIA 1736030 and OCE 15-59356 to G.F.S.; OCE 1756524 to S.T.W) and the Simons Foundation (#329108 to D.M.K. and A.E.W. and #811977 to S.T.W.). We thank Tina M. Weatherby and Marilyn F. Dunlap at the University of Hawai'i at Mānoa Biological Electron Microscopy

Facility for their assistance with transmission and scanning electron microscopy, Natalie Wallsgrove at the Stable Isotope Facility for her support, and the Hawaii Ocean Time-series program for their assistance with sample collection at Station ALOHA (NSF OCE 1756517 to D.M.K. and A.E.W.). The technical support and advanced computing resources from the University of Hawai'i Information Technology Services – Cyberin-frastructure, funded in part by NSF MRI 1920304, are gratefully acknowledged.

## Author contributions

C.R.S., S.T.W., M.C., Q.L., and G.F.S. contributed to experimental design. C.R.S. isolated and maintained both strains of *Epithemia* and conducted the light and electron microscopy analysis and phylogenetic analysis. C.R.S. and Q.L. conducted gene sequencing and qPCR. C.R.S. with input from K.A.T.-K. conducted the metagenomic analysis. S.T.W. and M.C. conducted the $N_2$ fixation measurements. R.S. contributed species descriptions and light microscopy imaging of cleaned diatom frustules. All authors, including C.R.S., S.T.W., M.C., R.S., Q.L., K.A.T.-K., A.E.W., D.M.K., J.P.Z., and G.F.S., contributed to the interpretation of results. C.R.S wrote the manuscript with input from all authors.

## Competing interests

The authors declare no competing interests.

## Additional information

**Peer review information** *Nature Communications* thanks Lucas Stal, Matt Ashworth, Rosalina Stancheva Hristova and the other anonymous reviewer(s) for their contribution to the peer review this work. Peer reviewer reports are available.

