## [Peer Review File · Nature Communications]

Reviewers' Comments:

Reviewer #1:

Remarks to the Author:

A novel, widespread diatom-diazotroph symbiosis in the sea

Review

This outstanding article provides the first evidence for the presence of nitrogen-fixing cyanobacterial endosymbions in two novel marine diatom species belonging to Rhopalodiaceae. This symbiosis is expected to occur in the marine *Epithemia*/*Rhopalodia* species, similarly to all freshwater species of both genera and was previously overlooked. The study provides significant original novel results from laboratory investigations of cultured diatoms, molecular phylogenetic data for the cyanobacterial endosymbiont, and its diatom host, and field data for distribution of the *nifH* sequences in environmental datasets.

Interesting novel observation is the loss of the endosymbiont after culturing the diatoms "for months in N-replete medium" (p.9, L. 310), although this symbiosis is considered obligatory for both organisms. I hope that authors could provide more specific details about this important new phenomenon, such as: how long exactly the diatoms were cultured in N-replete medium before the endosymbions disappear; is there observation how exactly the endosymbions were lost (probably during the cell division or sexual reproduction?); did you observe free endosymbiont cells in these cultures; how often did you check the cultures for the presence/absence of the endosymbionts? The Methods section could include explanation about the difference between "low-nitrogen medium" (p. 8, l. 273), "N-replete medium" (p.9, L. 310), "N-deplete medium" (p.9, L. 314).

Overall, the data for the presence of nitrogen-fixing cyanobacterial endosymbions in both diatom hosts is robust, valid and reliable. While the manuscript should ultimately be published, there are, in my opinion, some major adjustments needed before publication.

Improvements and additional evidence are recommended in the taxonomic description of the novel diatoms, which currently does not meet the phycology standards. Both diatom species need proper LM and SEM imaging and valid description, following the International Code of Nomenclature for algae, fungi, and plants, which is "a set of rules and recommendations that govern the scientific naming of all organisms traditionally treated as algae". The article is focused on the importance of the symbiotic relationship between coccoid cyanobacteria and their diatom host, but in fact the authors describe two novel algal species (diatoms) with cyanobacterial endosymbionts, and as such, the above code, botanical rules and phycological practice and terminology should be applied.

The main points that need to be addressed are outlined below:

1. The phylogenetic analysis shows that both novel species belong to family Rhopalodiaceae, but they are placed on separate branches, as authors state – "the diatom host phylogeny places *E. pelagica* and *E. catena* on separate branches of the revised *Epithemia* lineage" (p. 3, l. 82, 82), and "their endosymbionts reside on distinct phylogenetic branches" (p. 3, L. 80). The genetic distance between both species is also clearly manifested by their contrasting morphology. *E. pelagica* has single cells, strongly dorsiventral along the apical plane and wedge-shaped in the transapical plane, while *E. catena* forms long chains composed of elliptical cells in valve plane, which are wide rectangular in girdle view. The position of the raphe, which is important taxonomic feature, is along the dorsal margin in *E. pelagica* and median in *E. catena*. In summary, *E. pelagica* possess all characteristic features of the genus *Epithemia* (or former *Rhopalodia*), while *E. catena* does not, and may represent a novel genus.

According to Ruck et al. (2016), which provided the current molecular phylogeny of Rhopaladiales "in Rhopalodiales, cells are generally strongly dorsiventral along the apical plane and wedge-shaped in the transapical plane. Their canal raphe systems can be positioned along the dorsal (e.g., *Rhopalodia*) or ventral cell margin (e.g., *Epithemia*) or medially (e.g., *Tetralunata*). Indeed, sequence data are not available for *Tetralunata*, a recently described freshwater genus endemic to Indonesia by Hamsher et al. (2014), thus it was not considered in the formal analysis of Ruck et al. (2016), who decided to lump all the *Rhopalodia* taxa with all species of *Epithemia* within the

latter genus name. AlgaeBase is listing a few more diatom genera in Rhopalodiaceae.

2. The correct genus for *E. catena* needs to be determined, because the cell symmetry, raphe position, and the formation of long chains are not characteristic for *Epithemia*/*Rhopalodia*. In order to do so, the molecular data needs to be supplemented with proper high quality LM and SEM imaging of the frustules and detailed morphological description of the ultrastructure of the frustules.

3. LM imaging of the valve structure of both species is missing. The article contains only LM of living cells with protoplasts to illustrate the presence or absence of cyanobacterial endosymbiont, but these images are not appropriate for species identification, observation and description of frustule morphology. The diatom frustules should be cleaned from organic material and mounted in Naphrax. For each species, a plate with LM images illustrating the frustule gross morphology needs to be included (either in the main texts or as supplementary figures).

4. SEM imaging. The diatom material was not prepared for SEM. Authors used preparation method for TEM (transmission electron microscopy), which does not remove the organic content from the sample. As a result, the images of *E. catena* are not informative – the frustule ultrastructure (e.g., areolae, raphe, keel, etc.) are not clearly visible. Also, there are a lot of bacteria on the top of the diatoms (Fig. 1D – should be replaced). New high-quality SEM images of *E. catena*, focused on taxonomically important ultrastructure features must be obtained, because this is the only way to identify and describe this potentially novel diatom genus.

5. Both species need a valid description, including size ranges, frustule morphology and ultrastructure, etc. following the diatom taxonomy standards and the International Code of Nomenclature for algae, fungi, and plants. As an example, I provide article on description of a new diatom species applying all needed LM and SEM observations (Stancheva 2019). All names of algal genera and species, when mentioned for the first time in the text must have taxonomic authority.

6. The description of the cyanobacterial endosymbionts is incomplete. There is data only about average length. Data about length and width ranges is recommended. Also, the normal position of the endosymbionts in the host cells of *E. pelagica* is unclear. The endosymbionts are illustrated at the poles of the cells (Fig. 1C, Suppl. Fig. 1S1A), but clarification is needed on whether this is their normal position, or is that is a result of cell centrifugation or other manipulation, because the chloroplast is not in good shape in both cells as well. Typically the cyanobacterial endosymbionts in *Epithemia* are in the middle of the host cell.

I hope that the authors could provide the recommended additional morphological data and descriptions of both previously neglected diatom species, because it will improve our understanding about this important diatom-diazotroph symbiosis to the marine nitrogen cycle, as clearly demonstrated by the molecular findings in this article.

Rosalina Stancheva

References

- Ruck, E.C., Nakov, T., Alverson, A.J. & Theriot, E.C. (2016) Phylogeny, ecology, morphological evolution, and reclassification of the diatom orders Surirellales and Rhopalodiales. *Molecular Phylogenetics and Evolution* 103: 155–171.
<https://doi.org/10.1016/j.ympev.2016.07.023>
Stancheva, R. 2019. *Planothidium sheathii*, sp. nov, a new monoraphid diatom from rivers in California, USA. - *Phytotaxa* 393: 131-140

Reviewer #2:

Remarks to the Author:

This paper reports the discovery of a known freshwater diatom-diazotroph symbiosis in global ocean waters for the first time. This is a novel finding. This manuscript expands what is known about eukaryote-diazotroph symbioses globally and impacts understanding of the microbes behind global new N production in aquatic systems. The team expertly combines several methods to reveal the nature and potential biogeochemical impact of this symbiosis including isolation from the field, microscopy, quantification by qPCR, probing global metagenome and sequence databases in a sequence-based approach, and nitrogen fixation measurements. My comments overall are

minor but would make it easier to replicate this work by future researchers and would improve communication of some nuances of the results.

Minor Comments

Line 103-106: This statement needs revision as it does not stand alone without careful study of the supplemental figure. It is unclear until looking at the figure that the sediment trap data spans multiple years – supporting the “interannual presence” statement. Also, I am concerned that the number of reads detected in the metagenomic database are 1-3 for each sediment trap sample. This seems very low and needs further justification. How many reads would be found from any surface water contamination that could come from recovering the sediment traps? Could these reads be due to error during sequencing a similar sequence? How can that be ruled out? Further justification or explanation of the confidence of this observation are needed to claim that the symbiosis has a role in carbon export, especially. The role, if any seems very small if only 1-3 sequences are recovered from millions of sequences in each sample.

Line 107-111: Here is description of the diel pattern of N₂ fixation by the endosymbionts. I found the statement describing the results to be simplistic relative to the patterns revealed in the figure itself. A more careful description is needed – or perhaps some changes to the figure in case I am misunderstanding Figure 2. A respite in the first half of the dark period is noted. There also appears to be a period of no N fixation during a large portion of the light period. And the only dark period is measured at one timepoint early morning – which could be related to sunrise, if I understand the x-axis time scale and measurement binning correctly. Is it possible that the N fixation happening in early morning is ramping up for the day? I think this description needs to be better developed to be compatible with the observations presented for symbiont systems.

Line 123-124: Which genes do these primers and probes target?

Line 271: What were the criteria used for picking cells for isolation?

Line 314: Suggest that the number of copies recovered should be relative to the number of host cells extracted rather than the amount of DNA used. It is unclear if these are axenic cultures, so the amount of DNA recovered from the cultures could vary independently of the concentration of host cells.

Line 377-380: Here is a section of results on which size fractions the nifH sequences came from that should not be part of the methods but part of the main text.

Line 405: Please provide the sequence of the gene fragment used to create your standards to make it easy for other groups to build on, or replicate, your results.

Line 407-412: The limits of detection and reaction efficiency need to be reported for the qPCR assays presented and included in the Fig S3.

Figure 1n: It would be nice to include a line showing the cruise track of Tara so that we can see all the sample locations where you did not recover any reads of interest in addition to the stations where reads were recovered. I realize that's not possible for the hits in the SRA database, but should be for the Tara database. Same comment for Figure S2.

- Anne Thompson

Reviewer #3:

Remarks to the Author:

This paper describes a for the ocean novel diatom - cyanobacterial-derived diazotroph symbiosis. Two species of the pennate diatom *Epithemia* possess 1-2 endosymbionts. The authors showed convincingly that the endosymbionts are derived from cyanobacteria based on their 16S rRNA and nifH gene sequences. However, the endosymbionts did not seem to have (enough) chlorophyll and phycobilin pigments based on the absence of autofluorescence. The authors showed that N₂

fixation occurred during much of the day and night, although there was a daily pattern. This new symbiosis seemed to be widespread in the ocean but previously undetected because the diatom loses its endosymbiont when cultured in nitrogen replete medium.

The work is comprehensive and convincing and very well written. I have a few minor editorial comments:

L.30: I would use 'cultured' (in the meaning of taking into culture; whereas 'cultivated' has another meaning in the sense of adapting to). Also, as it is written here, the model system is cultivated. Rephrase. (e.g., a model system that allows the easy culturing of the organisms for the study of organelle evolution.). No culturing data are given in this paper. How long are cultures (diazotrophic) stable? Are they available and/or kept in culture collections?

L.38: bacteria and archaea (instead of 'prokaryotes'); compete with whom?

L.50: delete: 'have been shown to'

L.61: unicellular endosymbionts

L.64: lack

L.66: I don't think that this paper has proven that the endosymbiont has 'lost' this ability; it may just not be expressed.

L.107; 413; 433; 437: consider using 'daily' instead of 'diel'. The latter is not an English word and only used in biology to indicate a 24h period, which is in fact 'daily' (whereas 'diurnal' refers to changes during the daylight)

L.117: which fixes N₂ during the daytime

L.119: endosymbiont in roman

L.131: culturing

L.132: uncultured

L.269: 4,800 m

L.280: Epithemia in italics

L.282: light source?

L.356, 363, 379, 382: 1,000; 2,000; 4,000

L.389: rRNA

L.591: references

Manuscript: NCOMMS-21-07462-T

Responses to reviewer comments:

Reviewer #1:

Reviewer's comment: Interesting novel observation is the loss of the endosymbiont after culturing the diatoms "for months in N-replete medium" (p.9, L. 310), although this symbiosis is considered obligatory for both organisms. I hope that authors could provide more specific details about this important new phenomenon, such as: how long exactly the diatoms were cultured in N-replete medium before the endosymbionts disappear; is there observation how exactly the endosymbionts were lost (probably during the cell division or sexual reproduction?); did you observe free endosymbiont cells in these cultures; how often did you check the cultures for the presence/absence of the endosymbionts?

Response: We have addressed these questions by adding details to the *Methods* section (lines 375–379), as follows: "The loss of endosymbionts from *Epithemia* cultures (UHM3200 and UHM3210) was observed after propagating cells for four months in nitrogen-replete medium (K)³⁶, where approximately 5–10% of the culture was transferred to fresh medium about every two weeks. Observations were only made at the end of this growth period. Endosymbionts were not observed growing freely in these cultures..."

Reviewer's comment: The *Methods* section could include explanation about the difference between "low-nitrogen medium" (p. 8, l. 273), "N-replete medium" (p.9, L. 310), "N-deplete medium" (p.9, L. 314).

Response: "N-replete medium" was specified as K medium in the manuscript (lines 91 and 482). "Low-nitrogen medium" and "N-deplete medium" are the same medium, and we have clarified this by changing all instances of "nitrogen-deplete medium" to "low-nitrogen medium" to be consistent across the manuscript. Furthermore, when these terms are first used, we have included the nitrogen concentrations in the media, e.g., "N-replete medium (K medium¹⁸, with 5×10^{-5} M NH₄, 8.82×10^{-4} M NO₃..." (lines 74–75) and "low-N medium (5×10^{-8} M NH₄)" (lines 76–77).

Reviewer's comment: 2. The correct genus for *E. catena* needs to be determined, because the cell symmetry, raphe position, and the formation of long chains are not characteristic for *Epithemia/Rhopalodia*. In order to do so, the molecular data needs to be supplemented with proper high-quality LM and SEM imaging of the frustules and detailed morphological description of the ultrastructure of the frustules.

Response: The morphology of *E. catenata* is indeed different from all previously described *Epithemia*. However, at this time, we do not believe there is sufficient evidence to classify this species as a new genus. Our phylogenetic analyses of Rhopalodiaceae (Figs 1q, S14, S16, S17), show strong support for grouping *E. catena* with *Rhopalodia* sp. 3825.12 and *Rhopalodia* cf. *musculus* 23vi08.2C.1, and images for both strains show they do not share *E. catena*'s unique morphology (Ruck et al. 2016). Thus, *E. catena*'s morphology is not conserved on this branch of the tree and creating a new genus would lead to paraphyly within *Epithemia/Rhopalodia*. Thus, while it is possible the cluster that includes *E. catena* might be split into a separate genus at some point as more isolates and sequence data become available, we feel the conservative approach at this point is to maintain the proposed scope of the *Epithemia* genus for the reasons laid out in Ruck et al. (2016).

Reviewer's comment: 3. LM imaging of the valve structure of both species is missing. The article contains only LM of living cells with protoplasts to illustrate the presence or absence of cyanobacterial

endosymbiont, but these images are not appropriate for species identification, observation and description of frustule morphology. The diatom frustules should be cleaned from organic material and mounted in Naphrax. For each species, a plate with LM images illustrating the frustule gross morphology needs to be included (either in the main texts or as supplementary figures).

Response: LM imaging of cleaned frustules for both species has now been included in Supplementary Figs S1 a–h and S3.

Reviewer's comment: 4. SEM imaging. The diatom material was not prepared for SEM. Authors used preparation method for TEM (transmission electron microscopy), which does not remove the organic content from the sample. As a result, the images of *E. catena* are not informative – the frustule ultrastructure (e.g., areolae, raphe, keel, etc.) are not clearly visible. Also, there are a lot of bacteria on the top of the diatoms (Fig. 1D – should be replaced). New high-quality SEM images of *E. catena*, focused on taxonomically important ultrastructure features must be obtained, because this is the only way to identify and describe this potentially novel diatom genus.

Response: Additional SEM imaging of cleaned frustules has been performed for *E. catenata* and is provided in Supplementary Fig. S4 h–k. To obtain the most ultrastructural information possible for this unusual new species, we also performed TEM imaging of the same cleaned frustules, provided in Supplementary Fig. S5.

Reviewer's comment: 5. Both species need a valid description, including size ranges, frustule morphology and ultrastructure, etc. following the diatom taxonomy standards and the International Code of Nomenclature for algae, fungi, and plants. As an example, I provide article on description of a new diatom species applying all needed LM and SEM observations (Stancheva 2019). All names of algal genera and species, when mentioned for the first time in the text must have taxonomic authority.

Response: Dr. Rosalina Stancheva has joined our study as a new coauthor. She has contributed thorough and detailed species descriptions for both *E. pelagica* and *E. catenata*, which are authoritative and follow the highest standards for diatom taxonomy. These species descriptions are provided at the beginning of the Supplementary Information document and are referenced in lines 55–58 of the manuscript, as follows: “Here we report the discovery of two novel species of oceanic, endosymbiont-bearing, rhopalodiacean diatoms, *Epithemia pelagica* Schvarcz, Stancheva, Steward, sp. nov. and *Epithemia catenata* Schvarcz, Stancheva, Steward, sp. nov. (full descriptions in Supplementary Information).”

Reviewer's comment: 6. The description of the cyanobacterial endosymbionts is incomplete. There is data only about average length. Data about length and width ranges is recommended. Also, the normal position of the endosymbionts in the host cells of *E. pelagica* is unclear. The endosymbionts are illustrated at the poles of the cells (Fig. 1C, Suppl. Fig. 1S1A), but clarification is needed on whether this is their normal position, or is that is a result of cell centrifugation or other manipulation, because the chloroplast is not in good shape in both cells as well. Typically the cyanobacterial endosymbionts in *Epithemia* are in the middle of the host cell.

Response: We have added average width data to our description of the endosymbionts in the main text. We also clarified the location of the endosymbionts within the host cells and added two new panels to Fig. 1 to illustrate this (Fig. 1 n,o). The modified description (lines 65–69) has been changed to the following: “Both species typically foster 1–2 unicellular endosymbionts per cell, which have average cell dimensions of $2.9 \times 2.4 \mu\text{m}$ for *E. pelagica* (Fig. 1c, h, n) and $4.0 \times 2.5 \mu\text{m}$ for *E. catenata* (Fig. 1f, k, o) and tend to be centrally located next to the host cell's nucleus (Fig. 1n, o).”

The effect of our osmotic shock technique on endosymbiont location has been described in the *Methods* section (lines 364–367), as follows: “While this is a simple technique for detecting and visualizing endosymbionts (Fig. 1c, f), it does not accurately represent the natural location of endosymbionts within the host cells, as seen when compared to fixed cell preparations for epifluorescence microscopy (Fig. 1n, o).” We have similarly added the following clarification to the caption of Fig. 1 (lines 652–655), “The osmotic shock treatment (c, f) disrupts host cell contents and displaces the endosymbionts. The natural intracellular location of endosymbionts is illustrated in n and o”

Reviewer #2:

Reviewers comment: Line 103-106: This statement needs revision as it does not stand alone without careful study of the supplemental figure. It is unclear until looking at the figure that the sediment trap data spans multiple years – supporting the “interannual presence” statement. Also, I am concerned that the number of reads detected in the metagenomic database are 1-3 for each sediment trap sample. This seems very low and needs further justification. How many reads would be found from any surface water contamination that could come from recovering the sediment traps? Could these reads be due to error during sequencing a similar sequence? How can that be ruled out? Further justification or explanation of the confidence of this observation are needed to claim that the symbiosis has a role in carbon export, especially. The role, if any seems very small if only 1-3 sequences are recovered from millions of sequences in each sample.

Response: Reviewer #2 makes a number of comments in this paragraph

With regards to the comments about surface water contamination, it is not possible that the samples were contaminated with surface water during recovery of the sediment traps, because the 21 collection cups mounted on the rotating carousel are only open for collecting sinking particles for a predetermined time (in this instance, 12, 10, or 8 days). Before and after this 8–12 day period of sample collection, the cups are closed. We have amended the *Methods* section to include this information, and lines 476–480 now read “The presence of *EpSB* and *EcSB nifH* sequences was examined in metagenomes prepared from sinking particles collected at 4,000 m depth at Station ALOHA^{22,23} (Supplementary Table S4). The sinking particles were collected during intervals of 12, 10, and 8 days during 2014, 2015, and 2016, respectively, using a McLane sediment trap equipped with a 21-sample bottle carousel.”

We agree with Reviewer #2 that the text describing *E. pelagica*'s potential role in carbon export needed to be changed and we have instead emphasized the sediment trap data as evidence for a sustained population of *E. pelagica* in the surface waters of the North Pacific. Lines 107–115 have been revised as follows: “At Station ALOHA, where the isolates were obtained from, quantitative PCR measurements of *E. pelagica* symbioses indicated host LSU and *EpSB nifH* gene copies as high as $275 \pm 128 \times 10^3 \text{ L}^{-1}$ and $10 \pm 3 \times 10^3 \text{ L}^{-1}$, respectively (Supplementary Fig. S20a). Identical *EpSB nifH* sequences were also detected in metagenomes constructed from sinking particles collected at 4,000 m depth at Station ALOHA over a 3-year period (Supplementary Fig. S20b)^{22,23}. The recurring presence of *EpSB* sequences in sinking particles collected in the bathypelagic zone of the water-column is strong evidence for a sustained population in the surface waters of the North Pacific Subtropical Gyre.”

With regards to whether these *EpSB nifH* reads could have resulted from sequencing error, we addressed this by reanalyzing the sediment trap metagenomes after trimming low-quality bases from metagenomic reads. Leading and trailing bases with PHRED quality scores <20 (i.e., <99% base call accuracy) were trimmed, and only sequences >100 bp were retained. This resulted in a slight increase in the total number of identified *EpSB* reads (143 reads vs 137 reads). This is expected, since the Illumina NextSeq platform (v2 chemistry) used in the sequencing of the sediment trap metagenomes has a very low error rate (Manley et al. 2016, doi: 10.7171/jbt.16-2704-002). In cases where errors do arise, there is a higher probability that these random sequencing errors would result in sequences that diverge from the few known representative sequences (such as *EpSB*). We have employed a very strict criterion of requiring

100 percent nucleotide identity to *EpSB*, over the entirety of the Illumina sequence read. Given this strict criterion, sequencing errors would more likely lead to an underestimation of true *EpSB*-like *nifH* sequences in the datasets (e.g., a single error would render the sequence <100% identical to our *EpSB* reference sequences, and this sequence would therefore not be counted in our analysis).

However, reanalyzing the data in this way helped reveal a flaw in our original analysis. It was discovered that the NCBI SRA Toolkit that was originally used to analyze the environmental metagenomes does not always reliably download the target database. This leads to incomplete and partial analysis of the target databases. To remedy this issue, all SRA Toolkit analyses were re-run using a database verification step. This did not affect the results of our biogeographic analysis of *EpSB* and *EcSB*, but it did result in the identification of many more *EpSB*-like sequences in the sediment trap metagenomes (up to 24 reads in a single metagenome, instead of the previous maximum of 3). The reanalyzed data is now shown in Supplementary Fig. S20b, and the *Methods* section has been updated accordingly.

Reviewer's comment: Line 123-124: Which genes do these primers and probes target?

Response: We have clarified this by revising lines 137 as follows: "...*nifH* primers and probes..." We also added citations for the specific primer/probe sequences compared in Supplementary Fig. S22.

Reviewer's comment: Line 271: What were the criteria used for picking cells for isolation?

Response: Micropipette isolation was used to establish clonal cell lines from already established unialgal cultures. No criteria were used for picking cells. In each case, multiple cells were picked and separated into individual wells for growth. The most healthy-looking cultures established from these single cells were retained.

Reviewer's comment: Line 314: Suggest that the number of copies recovered should be relative to the number of host cells extracted rather than the amount of DNA used. It is unclear if these are axenic cultures, so the amount of DNA recovered from the cultures could vary independently of the concentration of host cells.

Response: Yes. These cultures are not axenic and contain minor bacterial contaminants, and we do not attempt to estimate the proportion of DNA originating from these different sources. However, in all cases, the DNA was extracted from healthy cultures, and the majority of DNA presumably originated from the diatom cells.

Reviewer's comment: Line 377-380: Here is a section of results on which size fractions the *nifH* sequences came from that should not be part of the methods but part of the main text.

Response: This sentence has been moved to the main text, at lines 99–102.

Reviewer's comment: Line 405: Please provide the sequence of the gene fragment used to create your standards to make it easy for other groups to build on, or replicate, your results.

Response: The gBlocks Gene Fragments sequences have been provided in Supplementary Table S5.

Reviewer's comment: Line 407-412: The limits of detection and reaction efficiency need to be reported for the qPCR assays presented and included in the Fig S3.

Response: These details have been added to the Methods section (lines 511–512), as follows: “Reaction efficiencies were 104.23% and 95.15% for the LSU and *nifH* genes, respectively.”

We did not determine the limit of detection for these qPCR assays, and we have now stated this at lines 512–513 as follows: “The limit of detection for these assays were not empirically determined.”

Reviewer's comment: Figure 1n: It would be nice to include a line showing the cruise track of Tara so that we can see all the sample locations where you did not recover any reads of interest in addition to the stations where reads were recovered. I realize that's not possible for the hits in the SRA database, but should be for the Tara database. Same comment for Figure S2.

Response: We have modified Fig. 1p and Supplementary Fig. S19 to indicate Tara Oceans stations that were included in the MATOU analysis. However, we did not mark the locations of analyzed SRA databases, since sample collection and processing methods vary between different SRA projects. For some SRA projects, the failure to detect *Epithemia* SB *nifH* could be solely caused by methodological differences, unrelated to the abundance of *Epithemia* SB at those locations.

Reviewer's comment: Line 107-111: Here is description of the diel pattern of N₂ fixation by the endosymbionts. I found the statement describing the results to be simplistic relative to the patterns revealed in the figure itself. A more careful description is needed – or perhaps some changes to the figure in case I am misunderstanding Figure 2. A respite in the first half of the dark period is noted. There also appears to be a period of no N fixation during a large portion of the light period. And the only dark period is measured at one timepoint early morning – which could be related to sunrise, if I understand the x-axis time scale and measurement binning correctly. Is it possible that the N fixation happening in early morning is ramping up for the day? I think this description needs to be better developed to be compatible with the observations presented for symbiont systems.

Response: Reviewer #2 highlights that a more careful description of the diel patterns of N₂ fixation is required. Specifically, Reviewer #2 makes two comments that both broadly refer to the impact of light availability on N₂ fixation and the occurrence of N₂ fixation in conjunction with the lights either being switched off (7pm) or on (7am). The first comment by Reviewer #2 states that “there also appears to be a period of no N₂ fixation during a large portion of the light period”. We believe Reviewer #2 is referring to the decrease in H₂ that occurs 2-3 h prior to the onset of the dark period for *Epithemia pelagica* (Figure 2a). In this instance, it is our interpretation that the decrease in N₂ fixation is preceding the onset of dark period, rather than an absence of N₂ fixation during the light period. This observation is supported by the ¹⁵N₂ measurements which is also plotted in Figure 2A. A decrease in N₂ fixation prior to the lights being switched off is a fairly common observation in other microorganisms that fix N₂ during the daytime (e.g., *Hemiaulus* as shown by Pyle et al. 2020, doi:10.7717/peerj.10115; and *Trichodesmium* as shown by Wilson et al. 2010, doi: 10.3354/ame01407). However, it should be noted that there have also been a number of observations where a more abrupt cessation of N₂ fixation occurs when the lights are switched off and not beforehand. Indeed, this is observed in our study for *Epithemia catena* (Figure 2b) and also for *Trichodesmium* (e.g., Rodriguez and Ho, 2014, doi: 10.1038/srep04445). We have revised the text to reflect this variability in the two *Epithemia* strains (see below). The second comment by Reviewer #2 relates to the increase in N₂ fixation during the night period. Reviewer #2 highlights that this occurs “early morning” and N₂ fixation might be “rampng up for the day”. For both strains of *Epithemia*, N₂ fixation is actually increasing between midnight and

1am, which is 5-6 hours earlier than would be expected if this were associated with the onset of light (lights came on at 7am). This is why the original manuscript text states that N₂ fixation occurred during the day and the night period at equal measure. Ultimately, these comments by Reviewer #2 indicate that our description and interpretation of these daily patterns needed to be improved and we have revised lines 116–131 as written below.

“The daily patterns of N₂ fixation in *E. pelagica* and *E. catenata* endosymbionts are distinct from other pelagic diazotrophs. In general, N₂ fixation by marine cyanobacteria occurs during either the day or the night²⁵. For both *EpSB* and *EcSB*, N₂ fixation occurred during the day and night (Fig. 2). During the day, N₂ fixation ceases either a few hours prior to the end of the light period (*E. pelagica*; Fig. 2a) or in conjunction with the lights being switched off (*E. catenata*; Fig. 2b). For both strains, N₂ fixation was subsequently undetected for the first 6 h of the dark period and resumed just after midnight. Overall, *EpSB* and *EcSB* are able to fix N₂ for a much longer period of time during a day-night cycle than other marine diazotrophs, especially the unicellular *Crocospaera subtropica*^{26,27}, which is the closest free-living relative of rhopalodiacean endosymbionts (Fig. 1r). *C. subtropica* synthesizes its carbohydrates during the day and respire them at night to fuel nitrogenase, while the evolutionary transition to an endosymbiont has enabled the *EpSB* and *EcSB* spheroid bodies to perform N₂ fixation during the day which is most likely fueled by metabolism of the host cell¹⁴. A similar evolutionary transition is hypothesized to have occurred for UCYN-A which fixes N₂ during the daytime²⁸.”

Finally, please note that Figure 2 has been revised according to Nature editorial guidelines with the bar charts now replaced by individual data plots for the ¹⁵N₂ assimilation values. All of the data have been included in the Supplementary Material as Table S6.

Reviewer # 3

Reviewer’s comment: L.30: I would use ‘cultured’ (in the meaning of taking into culture; whereas ‘cultivated’ has another meaning in the sense of adapting to). Also, as it is written here, the model system is cultivated. Rephrase. (e.g., a model system that allows the easy culturing of the organisms for the study of organelle evolution.).

Reviewer’s comment: L.131: culturing

Reviewer’s comment: L.132: uncultured

Response: We have changed all variations of “cultivate” to “culture”, as suggested. We have also rephrased “...easily cultivated model systems” to “...easily cultured model organisms” for grammatical accuracy (line 32).

Reviewer’s comment: No culturing data are given in this paper. How long are cultures (diazotrophic) stable?

Response: The following sentence has been added to the *Methods* section (lines 344–345) to clarify the stability of these cultures: “All *E. pelagica* and *E. catena* symbioses were stable under these medium and incubation conditions.”

Reviewer’s comment: Are they available and/or kept in culture collections?

Response: The following sentence has been added to the *Data availability* section (lines 551–552): “*E. pelagica* and *E. catena* cultures are maintained in culture at the University of Hawai’i at Mānoa and are available upon request.”

Reviewer's comment: L.38: bacteria and archaea (instead of 'prokaryotes'); compete with whom?

Response: "Prokaryotes" has been changed to "bacteria and archaea". We have also removed the word "compete" and rephrased this sentence (lines 40–41) for clarity as follows: "...some eukaryotes have adapted to low-nitrogen concentrations in the oligotrophic ocean by establishing mutualistic symbioses with diazotrophic bacteria⁴."

Reviewer's comment: L.50: delete: 'have been shown to'

Response: This phrase has been deleted.

Reviewer's comment: L.61: unicellular endosymbionts

Response: We have added "unicellular" for clarity.

Reviewer's comment: L.64: lack

Response: We have changed "lacked" to "lack".

Reviewer's comment: L.66: I don't think that this paper has proven that the endosymbiont has 'lost' this ability; it may just not be expressed.

Response: The original text says that the loss of photosynthesis is implied, which we intended to be interpreted as only suggested, but not proven. We qualified the statement further using the word "may", to help emphasize this (line 71): "...implying that these endosymbionts may have lost their ability to photosynthesize..."

Reviewer's comment: L.107; 413; 433; 437: consider using 'daily' instead of 'diel'. The latter is not an English word and only used in biology to indicate a 24h period, which is in fact 'daily' (whereas 'diurnal' refers to changes during the daylight)

Response: We have changed all instances of "diel" to "daily," which we think will be more easily understood by a broad audience.

Reviewer's comment: L.117: which fixes N₂ during the daytime

Response: This sentence has been rephrased as suggested.

Reviewer's comment: L.119: endosymbiont in roman

Response: We have unitalicized "endosymbiont".

Reviewer's comment: L.280: Epithemia in italics

Response: This text has been changed to italics.

Reviewer's comment: L.282: light source?

Response: Information regarding the light source has been provided on lines 343–344: "...using cool white fluorescent bulbs"

Reviewer's comment: L.269: 4,800 m

Reviewer's comment: L.356, 363, 379, 382: 1,000; 2,000; 4,000

Response: Commas have been added to all of these number values.

Reviewer's comment: L.389: rRNA

Response: "rDNA" has been changed to "rRNA", as suggested.

Reviewer's comment: L.591: references

Response: The full references for the citations have been provided in the caption for Supplementary Fig. S21. We have also cited these studies in the *Methods* section at line 456.

Reviewers' Comments:

Reviewer #2:

Remarks to the Author:

The authors have addressed my concerns and I think the manuscript is ready for publication.

Reviewer #4:

Remarks to the Author:

This revised manuscript identifies two new marine diatom-diazotroph symbioses, documenting the morphology, molecular systematics, distribution and some aspects of physiology of the diatom hosts and endosymbionts.

Based on the previous reviews and the completeness of the responses to those reviews, I assume I've been asked to review this revision to specifically to address the diatom taxonomy aspect of this manuscript. The authors responded to reviewer criticism over the lack of proper description and documentation of these diatoms by bringing in a new author to address these issues. The revised version includes a well-written description of both taxa, excellent light and electron micrographs and molecular phylogenetic analyses from multiple markers to formally describe these new diatom taxa.

I would accept *Epithemia pelagica* without protest—the morphology conforms to the described concept of the genus (recently revised by Ruck et al. 2016) and the DNA sequence data agree with this interpretation of the taxonomy.

The second taxon—*Epithemia catenata*—is problematic. As the reviewer and authors point out, the morphology of this taxon is a significant departure from all other taxa in this genus. The ultrastructure and position of the canal raphe do not conform with *Epithemia*, nor any other taxon in the family Rhopalodiaceae. This taxon also lacks the internal costae and areolar shape and occlusions of *Epithemia*. The authors defend the placement of this taxon in the genus *Epithemia* solely on the DNA sequence data and the presence of cyanobacterial endosymbionts. While the authors do have multiple DNA markers, which all place this taxon sister to "*Rhopalodia*" (since transferred to *Epithemia*) sequences, it would appear that the authors are completely discounting the morphological evidence in favor of the DNA characters. This is problematic, as there are far more taxa described by morphology than DNA across the diatoms against which this taxon could be compared. Why are the authors willing to accept that the morphological characters are ambiguous or subject to homoplasy but not the DNA data?

I would recommend three additional steps for the authors to at least "cover their bases" with regard to the DNA data. 1) The ribosomal sequence data do not appear to be aligned by secondary structure. Diatom nuclear ribosomal sequences often have multiple (and in some cases, quite large) inserts and loops, which can be subject to different models of molecular evolution than the base pairs subject to compensatory base changes in the stem regions. Do the SSU and LSU data derive the same tree topology when applied to secondary structure models? 2) The taxon sampling in the molecular phylogeny does not include any Bacillariacean taxa (*Nitzschia* and its allies), which also possess fibulae supporting the raphe canal. If this taxon remains sister to *Epithemia*, this would support the hypothesis that this taxon belongs to the Rhopalodiaceae. 3) Test an alternative taxonomic hypothesis using the DNA sequence data—if this taxon is constrained to be sister to all other *Epithemia* taxa in the dataset (and potentially a new genus), is the resulting molecular phylogenetic tree significantly worse (SH-test or AU-test) than the current tree? If the tree topologies aren't significantly different, the DNA sequence data might have as much homoplasy as the authors assume the morphological characters possess.

Extraordinary claims require extraordinary evidence. While diatom taxonomy might not seem like an "extraordinary claim", the fact that *E. catenata* is so morphologically divergent from the rest of the genus is significant. The implication made by the taxonomic description in this manuscript is that the diagnosis of *Epithemia* becomes so wide that it would include taxa currently in the genus *Entomoneis* as well. I am not fundamentally opposed to the idea that "*Epithemia catenata*" might

be created solely as a placeholder until more data can be found to find a more appropriate taxonomy, but if the authors intend to do so I do not think it is unreasonable to ask them to critically test all their available data.

Manuscript: NCOMMS-21-07462-T

Responses to reviewer comments:

Reviewer #4:

This revised manuscript identifies two new marine diatom-diazotroph symbioses, documenting the morphology, molecular systematics, distribution and some aspects of physiology of the diatom hosts and endosymbionts.

Based on the previous reviews and the completeness of the responses to those reviews, I assume I've been asked to review this revision to specifically to address the diatom taxonomy aspect of this manuscript. The authors responded to reviewer criticism over the lack of proper description and documentation of these diatoms by bringing in a new author to address these issues. The revised version includes a well-written description of both taxa, excellent light and electron micrographs and molecular phylogenetic analyses from multiple markers to formally describe these new diatom taxa.

I would accept *Epithemia pelagica* without protest—the morphology conforms to the described concept of the genus (recently revised by Ruck et al. 2016) and the DNA sequence data agree with this interpretation of the taxonomy.

The second taxon—*Epithemia catenata*—is problematic. As the reviewer and authors point out, the morphology of this taxon is a significant departure from all other taxa in this genus. The ultrastructure and position of the canal raphe do not conform with *Epithemia*, nor any other taxon in the family Rhopalodiaceae. This taxon also lacks the internal costae and areolar shape and occlusions of *Epithemia*. The authors defend the placement of this taxon in the genus *Epithemia* solely on the DNA sequence data and the presence of cyanobacterial endosymbionts. While the authors do have multiple DNA markers, which all place this taxon sister to “*Rhopalodia*” (since transferred to *Epithemia*) sequences, it would appear that the authors are completely discounting the morphological evidence in favor of the DNA characters. This is problematic, as there are far more taxa described by morphology than DNA across the diatoms against which this taxon could be compared. **Why are the authors willing to accept that the morphological characters are ambiguous or subject to homoplasy but not the DNA data?**

Response: Homoplasy in genetic data is perhaps a concern when dealing with highly similar sequences that only differ by a few residues, but in our case, we are basing our results on five maker genes, representing all major organellar genomes (nucleus, chloroplast, mitochondria) and including hundreds of variable bases. We are not aware of examples of genetic homoplasy to this extent.

I would recommend three additional steps for the authors to at least “cover their bases” with regard to the DNA data. 1) The ribosomal sequence data do not appear to be aligned by secondary structure. Diatom nuclear ribosomal sequences often have multiple (and in some cases, quite large) inserts and loops, which can be subject to different models of molecular evolution than the base pairs subject to compensatory base changes in the stem regions. **Do the SSU and LSU data derive the same tree topology when applied to secondary structure models?**

Response: We agree that it is worthwhile to investigate the results of trees based on alignments informed by secondary structure. To this end, we tested the use of SSU-ALIGN for the SSU rRNA gene and the SINA Aligner for both the SSU and LSU rRNA genes. We also tested a variety of post-alignment trimming strategies, such as no trimming, SSU-ALIGN's mask for low-confidence sites, SINA's positional variability filter, and trimAl's gappyout method. Maximum likelihood phylogenies were created using combinations of these alignment and trimming strategies. The phylogenies were then inspected to assess how well they maintained monophyly within the

previously described Rhopalodiales clade (e.g., was *Protokeelia* placed in this clade? Is the clade missing members? Does the clade contain non-Rhopalodiales members?) and how well they prevented separation of known conspecific strains (e.g., are the very closely related *E. pelagica* strains clustered together, regardless of their location on the tree?). These metrics were used to assess the accuracy of the alignments and the retention of a sufficient number of informative sites to resolve known relationships. Based on these metrics we chose an alignment and trimming strategy for both the SSU and LSU rRNA genes, which were then used for constructing two new gene trees: Supplementary Information Figures S14 and S16. The Methods section (lines 482–490) was updated accordingly as follows:

“rRNA gene phylogenies were also inferred using sequences aligned according to the global SILVA alignment for SSU and LSU genes using SINA⁵⁴, which were either left untrimmed in the case of the LSU gene or trimmed to remove highly variable positions (SINA’s “012345” positional variability filter) and gappy positions (trimAL v1.2, gappyout method) in the case of the 18S rRNA gene. These trimming strategies were selected based on their ability to maximize the monophyly of the previously described Rhopalodiales clade and minimize the separation of known conspecific strains, such as the strains of *E. pelagica* described here.”

The topologies of the trees using rRNA-specific aligners are not significantly different from the trees based on de novo alignment using MAFFT. In all cases, *E. catenata* is placed within Rhopalodiales and among other strains of *Epithemia* and *Rhopalodia*. While this was worthwhile to confirm, it is not unexpected, since the inclusion of *E. catenata* within Rhopalodiales is consistent with the three non-ribosomal gene phylogenies (*cob*, *psbC*, *rbcL*).

2) The taxon sampling in the molecular phylogeny does not include any Bacillariacean taxa (*Nitzschia* and its allies), which also possess fibulae supporting the raphe canal. If this taxon remains sister to *Epithemia*, this would support the hypothesis that this taxon belongs to the Rhopalodiaceae.

Response: We previously investigated the phylogenetic placement of *E. catenata* when including a much broader set of taxa, including diatom sequences from the orders Bacillariales (*Nitzschia*, *Pseudo-nitzschia*), Cymbellales (*Didymosphenia*), Naviculales (*Amphiprora*, *Navicula*, *Pinnularia*), and Thalassiophysales (*Amphora*, *Halamphora*, *Thalassiophysa*). For all our five tested genes, *E. catenata* was consistently placed within Rhopalodiales. Because sequences from these other taxa diverge greatly from *E. catenata* and could adversely affect the accuracy of the sequence alignment, we restricted our gene phylogenies to all known representatives of the orders Surirellales and Rhopalodiales.

We have updated our methods section (lines 495–499) to indicate that these tests were performed, as follows:

“Given *E. catenata*’s unusual morphology, test trees were inferred with the inclusion of diatom sequences from orders Bacillariales (*Nitzschia*, *Pseudo-nitzschia*), Cymbellales (*Didymosphenia*), Naviculales (*Amphiprora*, *Navicula*, *Pinnularia*), and Thalassiophysales (*Amphora*, *Halamphora*, *Thalassiophysa*); however, *E. catenata* was consistently placed within Rhopalodiales, and these trees were not pursued further.”

We also updated our gene phylogenies with symbols marking the top BLAST hits for *E. catenata* and *E. pelagica*, to indicate the taxa to which they share highest sequence similarity. We hope this will further convince readers that *E. catenata* is being analyzed in the appropriate phylogenetic context.

3) Test an alternative taxonomic hypothesis using the DNA sequence data—if this taxon is constrained to be sister to all other *Epithemia* taxa in the dataset (and potentially a new genus), is the

resulting molecular phylogenetic tree significantly worse (SH-test or AU-test) than the current tree? If the tree topologies aren't significantly different, the DNA sequence data might have as much homoplasy as the authors assume the morphological characters possess.

Response: We performed SH and AU tests, as requested. The tests were performed on all individual gene tree alignments, including the new rRNA gene-specific alignments, using IQ-TREE 2. Tests were performed for five different topological constraints: (1) the inclusion of *E. catenata* within genus *Epithemia*; (2) grouping *E. catenata* with a subset of closely related *Epithemia/Rhopalodia*, which together form a sister clade to the remaining *Epithemia/Rhopalodia*—this clade was informed by our phylogenetic analyses; (3) placing the clade containing *E. catenata* and the subset of closely related *Epithemia/Rhopalodia* outside of the rest of *Rhopalodiales*; (4) placing *E. catenata* within *Rhopalodiales* and sister to all *Epithemia/Rhopalodia*; and (5) placing *E. catenata* outside of all *Rhopalodiales*.

The results provide greatest support for the inclusion of *E. catenata* within the *Epithemia* genus (whether grouped with closely related strains or not), and there was less support for *E. catenata* as a separate sister group to all currently described *Epithemia/Rhopalodia* (e.g., if *E. catenata* by itself represented a new genus). The results of these tests are consistent with our extensive phylogenetic analyses. This is perhaps expected, since the branches grouping *E. catenata* with other strains of *Epithemia* were estimated with high support values, and in most cases, these branches had complete support (posterior probability = 1).

The SH and AU test results are summarized in a new table, Supplementary Information Table S1, and the Methods section has been updated (lines 491–494) to reflect these new analyses as follows:

“To investigate the level of support for constrained tree topologies placing *E. catenata* within or outside of the genus *Epithemia* and family *Rhopalodiaceae*, SH⁵⁵ and AU⁵⁶ statistical tests were performed in IQ-TREE 2⁵⁷ (implementing ModelFinder⁵⁸) using all alignments from the individual gene trees (Supplementary Table S1).”

Extraordinary claims require extraordinary evidence. While diatom taxonomy might not seem like an “extraordinary claim”, the fact that *E. catenata* is so morphologically divergent from the rest of the genus is significant. The implication made by the taxonomic description in this manuscript is that the diagnosis of *Epithemia* becomes so wide that it would include taxa currently in the genus *Entomoneis* as well. I am not fundamentally opposed to the idea that “*Epithemia catenata*” might be created solely as a placeholder until more data can be found to find a more appropriate taxonomy, but if the authors intend to do so I do not think it is unreasonable to ask them to critically test all their available data.

Response: We agree that, given the unique morphology *E. catenata*, claims to its taxonomic placement should be well supported. It is our opinion that we have gone above and beyond in our investigation of this species, including extensive microscopy (light, scanning electron, and transmission electron) and robust multi-gene phylogenetic tree inference and hypothesis testing using genes from three different organellar genomes (nuclear, mitochondrial, and chloroplast).

While the morphology of *E. catenata* is a significant departure from other taxa in the genus *Epithemia*, it also does not conform to the morphology of any other described genus. Thus, the generic placement of this species cannot be made based on morphology alone. It came to our attention that *E. catenata* shares important morphological characteristics with *Nitzschia nienhuisii* F. A. S. Sterrenburg and F. J. G. Sterrenburg (Sterrenburg & Sterrenburg, 1990), and these species are very likely congeneric. However, the morphology of *N. nienhuisii* does not conform to the genus *Nitzschia*, and its morphology has baffled taxonomists, leading them to doubt *N. nienhuisii*'s generic placement in *Nitzschia*. No molecular data currently exists for *N. nienhuisii*, and its taxonomy is considered operative—a placeholder, so to speak.

The cases of *E. catenata* and *N. nienhuisii* are precisely those that must rely on molecular phylogenetics to inform their taxonomy. For *E. catenata*, all our extensive phylogenetic analyses are robustly consistent with including *E. catenata* within the genus *Epithemia*, as currently defined by Ruck et al. (2016). While assigning *E. catenata* (and *N. nienhuisii*) a new

genus name would simplify the descriptions of each genus and create a clean demarcation between these two different morphologies, it is not supported by either the multi-gene phylogenetics nor the SH and AU topology tests. Creating a new genus for *E. catenata* would directly contradict our extensive molecular analyses and would likely confuse readers. In agreement with the data we have in hand and which is presented in this manuscript, we believe the conservative approach is to place *E. catenata* within the genus *Epithemia*, until future data suggest otherwise, or until the genus *Epithemia* is restructured. In this sense, “*Epithemia catenata*” can be considered an operative placeholder.

We have modified the main text of the manuscript (lines 90–97) to acknowledge *E. catenata*'s morphological similarity to *N. nienhuissii*, as follows:

“*E. catenata*'s cell morphology differs significantly from the rest of Rhopalodiales²¹ but shares many characteristics with the tentatively classified *Nitzschia nienhuisii* F.A.S Sterrenburg & F.J.G. Sterrenburg²³, for which no molecular data exists. Spheroid body-like structures are visible in a previously captured photomicrograph of *N. nienhuisii* (Lobban 2015²⁴, Figure 89), suggesting that this species, which has been observed in the Pacific²⁴ and Atlantic²³ Oceans and Caribbean Sea^{25,26}, may also harbor endosymbionts.”

We have also modified the species description of *E. catenata* (Supplementary Information, lines 224–233) to describe *E. catenata*'s relationship to *N. nienhuissii* and provide proper context, as follows:

“*E. catenata* does, however, share important morphological characteristics with *Nitzschia nienhuisii* F. A. S. Sterrenburg and F. J. G. Sterrenburg, such as gross frustule symmetry, keel shape and position, chain-formation and lack of LM and SEM visible striation of the hyaline frustules (Sterrenburg & Sterrenburg 1990, LM figures 2-4; Lobban 2015, LM figures 89-92, SEM figures 93, 94; López-Fuerte et al. 2015, SEM figures 2 a, b, LM figures 2 c, d), suggesting that these two species may be congeneric. The initial description of *N. nienhuisii* acknowledges there is doubt with respect to *N. nienhuisii*'s generic ranking (Sterrenburg and Sterrenburg, 1990), and a later study employing SEM noted the baffling structure of *N. nienhuisii*, which unlike genus *Nitzschia* does not appear to possess fibulae on the keel (Lobban 2015). No molecular data currently exists for *N. nienhuisii*.”

Reviewer #4 also had these comments to the editor that they have authorized me to share:

If it were up to me, I would not formally describe this strain--give it an informal name rather than create more taxonomic problems among the diatoms. The inclusion of a completely different morphological character set into *Epithemia* has implications for both the genus and the family. If the authors want to create such a problem, I would ask that they be as critical of their DNA data as they are the morphological data.

Response: We are aware of the implications that classifying *E. catenata* as *Epithemia* has for both the genus and family. We would argue, however, that rather than creating a problem, we have merely identified a “problem” that has been there all along—i.e., the existence of novel frustule morphologies in *Epithemia* and Rhopalodiaceae. Following Ruck et al.'s (2016) analysis and reasoning, we are trying to avoid making things worse by introducing paraphyly in this clade.

Reviewers' Comments:

Reviewer #4:

Remarks to the Author:

I appreciate the efforts by the authors to address my concerns with the DNA sequence-based phylogeny. While I am still unsatisfied by the conclusions drawn, I cannot dispute the results based on these data and cannot reasonably object to the publication of this manuscript.

While I appreciate the effort to "avoid making things worse by introducing paraphyly in this [molecular] clade", I would still argue that paraphyly has well-and-truly been introduced into the systematics of the Rhopalodiaceae. At the very least, I appreciate that it was not a decision made lightly and that a complete framework was constructed around the taxonomic decision, which can be debated going forward.

...and oh, how I expect a spirited debate.